# A Geometric Analysis of PCA

**Ayoub El Hanchi**
University of Toronto &
Vector Institute
aelhan@cs.toronto.edu

**Murat A. Erdogdu**
University of Toronto &
Vector Institute
erdogdu@cs.toronto.edu

**Chris J. Maddison**
University of Toronto &
Vector Institute
cmaddis@cs.toronto.edu

## Abstract

What property of the data distribution determines the excess risk of principal component analysis? In this paper, we provide a precise answer to this question. We establish a central limit theorem for the error of the principal subspace estimated by PCA, and derive the asymptotic distribution of its excess risk under the reconstruction loss. We obtain a non-asymptotic upper bound on the excess risk of PCA that recovers, in the large sample limit, our asymptotic characterization. Underlying our contributions is the following result: we prove that the negative block Rayleigh quotient, defined on the Grassmannian, is generalized self-concordant along geodesics emanating from its minimizer of maximum rotation less than $\pi/4$.

## 1 Introduction

Principal Component Analysis (PCA) is a core method of machine learning and statistics, prized for its simplicity and consistently strong empirical performance across diverse tasks. In contrast, analyzing its theoretical performance is challenging: an explicit characterization of its asymptotic excess risk is known for the special case of Gaussian data [RW20], while non-asymptotic results on the excess risk are in general limited to upper bounds [Sha+05; BBZ07; Nad08; RW20].

Traditionally, two main approaches have been adopted to analyze PCA. In the first, it is treated as a plug-in estimator: the empirical covariance replaces the population one, and its principal components estimate the true ones. Matrix perturbation bounds [SS90], most famously the Davis-Kahan theorem [DK70; YWS15], are then used to control the error of PCA by the deviation of the empirical covariance from the population one. In the second approach, adopted in [Sha+05; BBZ07], PCA is viewed as an instance of empirical risk minimization, for which variants of the uniform convergence analysis apply. Unfortunately, neither of these approaches leads to provably accurate bounds.

The recent work of Reiss and Wahl [RW20] takes a different approach and analyzes the projector found by PCA directly, building on earlier work of Dauxois et al. [DPR82] who established its asymptotic normality. The excess risk bounds obtained therein are powerful enough that under Gaussian data and certain eigenvalue decay assumptions, they recover the leading term in the exact asymptotic expansion of the excess risk. In the general case, however, asymptotically tight bounds do not appear to be available in the existing literature.

In this paper, we take a different approach that allows us, among other things, to obtain such bounds: we view PCA as an M-estimator, and use tools from the theory of asymptotic statistics [Van00], and its accompanying non-asymptotic theory [e.g. OB21], to analyze its performance. From this point of view, PCA is similar to linear regression with ordinary least squares. A significant difference, and a major source of difficulty, resides in the nature of their respective search spaces: for linear regression, it is $\mathbb{R}^d$, whereas for PCA, it is the manifold of $k$-dimensional subspaces of $\mathbb{R}^d$ - the Grassmannian [e.g. EAS98]. We build extensively on the accessible expositions in [Bou23; BZA24] to overcome this difficulty.

39th Conference on Neural Information Processing Systems (NeurIPS 2025).

Our contributions are as follows. In Theorem 1, we establish a central limit theorem for the error of the principal subspace obtained by PCA, and use this result to obtain the asymptotic distribution of its excess risk under the reconstruction loss, all under a necessary moment assumption. We then establish, in Theorem 2, a non-asymptotic upper bound on the excess risk that, in the large-sample limit, accurately recovers our asymptotic characterization. At the heart of our analysis is the following key result (Proposition 1): we prove that the reconstruction risk is generalized self-concordant — in a sense analogous to the one introduced by Bach [Bac10] in his analysis of logistic regression - when restricted to geodesics originating from its global minimizer of maximum rotation less than $\pi/4$.

The rest of the paper is organized as follows. In Section 2 we formalize our problem and provide an overview of the Grassmann manifold, restricting ourselves to the objects needed to state our theorems. In Section 3 we characterize the asymptotic performance of PCA. In Section 4 we establish the self-concordance of the reconstruction risk. In Section 5 we provide a non-asymptotic bound on the error of PCA, and conclude with a discussion in Section 6. Proofs of our statements are provided in the Appendix.

## 2   Problem setup & Background

The goal of linear dimensionality reduction is to project high-dimensional data onto a lower dimensional subspace while preserving as much information about the original data as possible. Specifically, given i.i.d. data points $(X_i)_{i=1}^n$ in $\mathbb{R}^d$ and a choice of dimension $k \in [d]$, PCA finds an orthogonal projector $UU^T \in \mathbb{R}^{d \times d}$ onto a $k$-dimensional subspace of $\mathbb{R}^d$ such that the following empirical reconstruction error is as small as possible

$$\widetilde{R}_n(U) := \frac{1}{2n} \sum_{i=1}^n \|X_i - UU^T X_i\|_2^2. \tag{1}$$

Here $U \in \mathbb{R}^{d \times k}$ is a matrix whose columns form an orthonormal basis of the aforementioned subspace. Denote by $\Sigma_n := n^{-1} \sum_{i=1}^n X_i X_i^T$ the empirical covariance matrix, and fix an orthonormal basis $(u_{n,j})_{j=1}^d$ of eigenvectors of $\Sigma_n$, ordered non-increasingly according to their eigenvalues, with ties broken arbitrarily. The $d \times k$ matrix $U_n$ whose $j$-th column is $u_{n,j}$ is a minimizer of (1).

Typically, however, we care about the population reconstruction error of this projector. If $X$ is a random vector with the same distribution as that of the data points, this error is given by

$$\widetilde{R}(U) := \frac{1}{2} \mathrm{E}[\|X - UU^T X\|_2^2]. \tag{2}$$

Denote by $\Sigma := \mathrm{E}[XX^T]$ the population covariance matrix, and fix an orthonormal basis $(u_j)_{j=1}^d$ of eigenvectors of $\Sigma$, ordered non-increasingly according to their eigenvalues $(\lambda_j)_{j=1}^d$, with ties broken arbitrarily. The $d \times k$ matrix $U_*$ whose $j$-th column is $u_j$ is a minimizer of (2).

**A redundancy in the parametrization.**   The analysis of PCA is complicated by the fact that the map $U \mapsto UU^T$ is a redundant parametrization of the set of orthogonal projectors. Fortunately, this redundancy is well-structured: two such matrices $U, V$ represent the same projector (i.e. $UU^T = VV^T$) if and only if there exists an orthogonal matrix $Q \in \mathbb{R}^{k \times k}$ such that $V = UQ$. This defines an equivalence relation $\sim$ on the set $\mathrm{St}(d, k) := \{U \in \mathbb{R}^{d \times k} \mid U^T U = I_k\}$. The space of equivalence classes under this relation is known as the Grassmann manifold $\mathrm{Gr}(d, k) := \mathrm{St}(d, k)/\sim$. We denote a generic element in this space by $[U]$.

To gain some intuition about this abstract set, note that every element of $\mathrm{Gr}(d, k)$ can be identified with a $k$-dimensional subspace of $\mathbb{R}^d$ through the map that sends $[U]$ to the column space of $U$. Therefore for the sake of intuition we can think of $[U]$ as the column space of $U$. In the special case of $\mathrm{Gr}(3, 2)$, which we will use as a running example, we can visualize $[U]$ as a plane passing through the origin embedded in 3 dimensions.

Equipped with this new space, we define our final population and empirical risks by $R([U]) := \widetilde{R}(U)$ and $R_n([U]) := \widetilde{R}_n(U)$, and note that, with the definitions of $U_n$ and $U_*$ given above,

$$[U_*] \in \operatorname*{argmin}_{[U] \in \mathrm{Gr}(d,k)} R([U]), \qquad [U_n] \in \operatorname*{argmin}_{[U] \in \mathrm{Gr}(d,k)} R_n([U]). \tag{3}$$

To be precise, we will call PCA the abstract procedure that takes as input $(X_i)_{i=1}^n$ and outputs the subspace $[U_n]$, even though in practice it returns the representative $U_n$. In this paper, our goal is to understand the performance of this procedure in terms of the distribution of $X$, as measured both by how close $[U_n]$ is to $[U_*]$, and by the excess risk $R([U_n]) - R([U_*])$.

## 2.1 Background on the Grassmann manifold

Our analysis takes place on the Grassmannian $\mathrm{Gr}(d,k)$, which admits the structure of a Riemannian manifold. In this section, we give a high-level description of the objects needed to state our results. For a more rigorous yet accessible introduction, see for example [BZA24] and [Bou23, Chapter 9]. Throughout, we let $[U], [V] \in \mathrm{Gr}(d,k)$ and let $U, V \in \mathrm{St}(d,k)$ be corresponding representatives.

**Tangent space.** The tangent space of $\mathrm{Gr}(d,k)$ at $[U]$, denoted by $T_{[U]}\mathrm{Gr}(d,k)$, is a vector space of dimension $k(d-k)$. Our results are stated in terms of a more concrete, yet equivalent, set

$$H_U := \left\{ \Delta \in \mathbb{R}^{d \times k} \mid U^T \Delta = 0 \right\}. \tag{4}$$

For our purposes, $H_U$ is easier to work with and there exists a canonical, invertible, linear map[1] $\mathrm{lift}_U : T_{[U]}\mathrm{Gr}(d,k) \to H_U$ that lifts tangent vectors $\xi \in T_{[U]}\mathrm{Gr}(d,k)$ into $H_U$. The elements of $H_U$ can be thought of concretely as "velocities" in the sense that the basis $U$ with "velocity" $\mathrm{lift}_U(\xi)$ moves infinitesimally to "$U + \epsilon\,\mathrm{lift}_U(\xi)$" while remaining an orthonormal basis. By requiring that $\mathrm{lift}_U(\xi)$ has columns in the orthogonal complement of $U$, we are guaranteed that the subspace $[U]$ changes when $U$ moves in the direction of $\mathrm{lift}_U(\xi)$.

The tangent space is equipped with an inner product $\langle \cdot, \cdot \rangle_{[U]}$ known as the Riemannian metric at $[U]$. To calculate $\langle \cdot, \cdot \rangle_{[U]}$, we lift tangent vectors into $H_U$ and apply the Frobenius inner product,

$$\langle \xi_1, \xi_2 \rangle_{[U]} = \langle \mathrm{lift}_U(\xi_1), \mathrm{lift}_U(\xi_2) \rangle_F, \tag{5}$$

for all $\xi_1, \xi_2 \in T_{[U]}\mathrm{Gr}(d,k)$.

**Geodesics and the exponential map.** Let $\xi \in T_{[U]}\mathrm{Gr}(d,k)$. The geodesic starting at $[U]$ in the direction $\xi$ is the curve $\gamma : [0,1] \to \mathrm{Gr}(d,k)$ with $\gamma(0) = [U]$ of constant velocity $\xi$. Intuitively, one may think of $\gamma(t)$ as the straight line "$[U] + t\xi$", in the sense that $\gamma$ is a curve with zero acceleration, properly defined. $\gamma(t)$ can be calculated with the SVD $\mathrm{lift}_U(\xi) = PSQ^{T}$[2] via the identity

$$\gamma(t) := [UQ\cos(tS)Q^T + P\sin(tS)Q^T]. \tag{6}$$

The exponential map at $[U]$ in the direction $\xi$ is then defined by $\mathrm{Exp}_{[U]}(\xi) = \gamma(1)$. Returning to our running example, $\mathrm{Gr}(3,2)$, these geodesics correspond to "constant velocity" rotations of the plane $[U]$ along the pitch or roll axes, as specified by the directions of the columns of $\mathrm{lift}_U(\xi)$, and the extent of rotation between $[U]$ and $\mathrm{Exp}_{[U]}(\xi)$ depends on the magnitudes of the columns of $\mathrm{lift}_U(\xi)$.

**Principal angles and Riemannian distance.** The $j$-th principal angle $\theta_j([U],[V]) \in [0, \pi/2]$ is defined by $\cos(\theta_j([U],[V])) = s_j$ where $s_j$ is the $j$-th largest singular value of $U^T V$. These angles generalize the notion of angles between lines to angles between subspaces, and they measure the magnitude of the most efficient rotation that aligns $[U]$ with $[V]$. For $\mathrm{Gr}(d,k)$, the principal angles give us an explicit expression for the Riemannian distance between $[U]$ and $[V]$, which, properly defined, is the length of a shortest curve connecting $[U]$ and $[V]$,

$$\mathrm{dist}^2([U],[V]) := \sum_{j=1}^{k} \theta_j^2([U],[V]). \tag{7}$$

**Logarithmic map.** Where well-defined, the logarithmic map at $[U]$ evaluated at $[V]$ is the inverse of the exponential map, *i.e.*, $\mathrm{Exp}_{[U]}(\mathrm{Log}_{[U]}([V])) = [V]$. It can be thought of as "$[V] - [U]$". For $\mathrm{Gr}(d,k)$, the logarithmic map can be calculated by $\mathrm{lift}_U(\mathrm{Log}_{[U]}([V])) = (P\arctan(S)Q^T)$ where $(I - UU^T)V(U^T V)^{-1} = PSQ^T$ is a SVD. This map is only well-defined for $\theta_k([U],[V]) < \pi/2$. The singular values of $\mathrm{lift}_U(\mathrm{Log}_{[U]}([V]))$ are the principal angles, and hence the following holds $\|\mathrm{lift}_U(\mathrm{Log}_{[U]}([V]))\|_F = \mathrm{dist}([U],[V])$, see also [AV24]. In $\mathrm{Gr}(3,2)$, the logarithmic map at $[U]$ evaluated at $[V]$ gives us the most efficient rotation that transforms the plane $[U]$ into $[V]$.

---

[1] The inverse of the differential of the quotient map $U \mapsto [U]$ at $U$ restricted to $H_U$.
[2] $P \in \mathbb{R}^{d \times k}, S \in \mathbb{R}^{k \times k}, Q \in \mathbb{R}^{k \times k}$.

## 3 Asymptotic characterization

Before stating our first main result, we briefly recap our notation. The empirical and population covariance matrices are denoted by $\Sigma_n = n^{-1} \sum_{i=1}^{n} X_i X_i^T$ and $\Sigma = \mathrm{E}[XX^T]$, $(u_{n,j})$ is an orthonormal basis of eigenvectors of $\Sigma_n$ ordered non-increasingly according their corresponding eigenvalues, and $(u_j)$ is an orthonormal basis of eigenvectors of $\Sigma$ ordered non-increasingly according to their corresponding eigenvalues $(\lambda_j)$. $U_n \in \mathrm{St}(d, k)$ is the matrix whose $j$-th column is $u_{n,j}$, and corresponds to the output of PCA, while $U_* \in \mathrm{St}(d, k)$ is the matrix whose $j$-th column is $u_j$.

We further define $U_*^{\perp} \in \mathrm{St}(d, d - k)$ to be the matrix whose $i$-th column is $u_{k+i}$. It is easy to verify that the map $\Gamma \mapsto U_*^{\perp} \Gamma$ for $(d - k) \times k$ matrices $\Gamma$ is linear and its image is $H_{U_*}$ as defined in (4). It is invertible and preserves the Frobenius inner product, so $H_{U_*}$ can be identified with $\mathbb{R}^{(d-k) \times k}$ through it. Finally, recall that the logarithm at $[U_*]$ of $[U_n]$ is only well-defined when all the principal angles between them are strictly less than $\pi/2$. In what follows, this logarithm can be defined arbitrarily when this condition fails - the validity of the statement is unaffected by this choice.

The following is the first main result of the paper.

**Theorem 1.** *Assume that $\lambda_k > \lambda_{k+1}$, $\mathrm{E}[\|X\|_2^2]$ is finite, and for all $i, s \in [d - k]$ and $j, t \in [k]$,*

$$\Lambda_{ijst} := \mathrm{E}[\langle u_{k+i}, X \rangle \langle u_j, X \rangle \langle u_{k+s}, X \rangle \langle u_t, X \rangle] \tag{8}$$

*is finite. Define $\delta_{ij} := \lambda_j - \lambda_{k+i}$. Then as $n \to \infty$, the following holds.*

- *Consistency:*
$$\mathrm{P}(\mathrm{dist}([U_n], [U_*]) > \varepsilon) \to 0,$$
  *for all $\varepsilon > 0$, where* dist *is the Riemannian distance given by (7).*

- *Asymptotic normality:*
$$\sqrt{n} \cdot \mathrm{lift}_{U_*}(\mathrm{Log}_{[U_*]}([U_n])) \xrightarrow{d} U_*^{\perp} G, \tag{9}$$
  *where $G$ is a mean zero $(d - k) \times k$ Gaussian matrix with $\mathrm{E}[G_{ij} G_{st}] = \Lambda_{ijst}/\delta_{ij}\delta_{st}$.*

- *Excess risk:*
$$n \cdot [R([U_n]) - R([U_*])] \xrightarrow{d} \frac{1}{2}\|H\|_F^2, \tag{10}$$
  *where $H$ is a mean zero $(d - k) \times k$ Gaussian matrix with $\mathrm{E}[H_{ij} H_{st}] = \Lambda_{ijst}/\sqrt{\delta_{ij}\delta_{st}}$.*

Under an eigengap and moment condition, Theorem 1 characterizes the performance of PCA in the large sample limit. The consistency statement says that, with enough data, the principal subspace found by PCA gets arbitrarily close to the true one under the Riemannian distance (7) with overwhelming probability. The asymptotic normality result refines this statement: it says that the fluctuations of PCA around the true principal subspace are asymptotically normal with the prescribed covariance structure. Finally, the last statement expresses the asymptotic distribution of the excess risk as the squared Frobenius norm of a Gaussian matrix.

**Relationship with existing work and assumptions.** The consistency result is a direct consequence of the Davis-Kahan theorem [DK70]. To the best of our knowledge, the asymptotic normality result in Theorem 1 is new. The finiteness of (8) is necessary, in the same way that finite variance is for the classical central limit theorem. Tripuraneni et al. [Tri+18] obtained a similar expression for the asymptotic variance of averaged Riemannian SGD on PCA, albeit under an unverified assumption. Dauxois et al. [DPR82] established the asymptotic normality of the Euclidean fluctuations of empirical projectors and eigenvectors - our result may be viewed as a Riemannian analogue of theirs. Under the eigengap condition, the excess risk bound in Theorem 1 extends the result of Reiss and Wahl [RW20, Proposition 2.14]. While their statement is restricted to Gaussian data, the underlying argument carries over directly to any distribution with finite fourth moments. Theorem 1 strengthens this result further by requiring only the finiteness of (8). A detailed discussion of the eigengap condition is deferred to Section 6; for now, we simply note that it is a mild assumption.

Our main interest is in the excess risk, as it directly measures how well PCA performs on the reconstruction task. Corollary 1 below offers a more interpretable version of the result in Theorem 1, and serves as a benchmark for our non-asymptotic analysis. To motivate it, we briefly digress.

Performance guarantees in machine learning are typically stated as: with probability at least $1 - \delta$, the excess risk is at most some quantity. While intuitive, the accuracy of such statements is hard to quantify: what is a "high-probability lower bound"? This ambiguity can be avoided by interpreting such statements as upper bounds on the $1 - \delta$ quantile of the excess risk. Specifically, recall that for a random variable $Z$, its $1 - \delta$ quantile, for $\delta \in [0, 1]$, is defined by

$$Q_Z(1 - \delta) := \inf\{t \in \mathbb{R} \mid P(Z \leq t) \geq 1 - \delta\}.$$

In words, this quantile describes the best upper bound on the random variable $Z$ that holds with probability at least $1 - \delta$. We may then make sense of a "high-probability lower bound" on the excess risk as a lower bound on its $1 - \delta$ quantile. The following corollary, a simple consequence of Gaussian concentration, gives matching upper and lower bounds on the asymptotic quantiles of the excess risk.

**Corollary 1.** *In the setting of Theorem 1, and for all $\delta \in [0, 0.1)$*

$$\lim_{n \to \infty} n \cdot Q_{\mathcal{E}_n}(1 - \delta) \asymp \sum_{i=1}^{d-k} \sum_{j=1}^{k} \frac{\mathrm{E}[\langle u_{k+i}, X \rangle^2 \langle u_j, X \rangle^2]}{\lambda_j - \lambda_{k+i}} + 2\tau^2 \log(1/\delta),$$

*where $\mathcal{E}_n = R([U_n]) - R([U_*])$ is the excess risk of PCA, $\tau^2 = \sup_{\|A\|_F=1} \mathrm{E}[\langle A, H \rangle_F^2]$, and $a \asymp b$ means that $cb \leq a \leq Cb$ for some $C, c > 0$. Here $C = 1$ and $c = 1/64$ are valid choices.*

To interpret this statement, it will be useful to introduce the following definition. For a random matrix $W \in \mathbb{R}^{d \times k}$, we define its covariance operator to be the linear map $\mathrm{Cov}(W) : \mathbb{R}^{d \times k} \to \mathbb{R}^{d \times k}$ given by $\mathrm{Cov}(W)[A] := \mathrm{E}[\langle W, A \rangle_F W]$. It is positive semi-definite, i.e. $\langle A, \mathrm{Cov}(W)[A] \rangle_F \geq 0$ for all $A$, and so has non-negative eigenvalues, whose sum is the trace of $\mathrm{Cov}(W)$, which equals $\mathrm{E}[\|W\|_F^2]$.

Corollary 1 then says that for sufficiently small failure probability $\delta$, the $1 - \delta$ asymptotic quantile of the excess risk is equivalent, up to explicit constants, to a sum of two terms. The first is $\mathrm{E}[\|H\|_F^2]$ which equals the trace of $\mathrm{Cov}(H)$ - that is the sum of its eigenvalues. It admits an explicit expression in terms of (i) a second-order covariance between pairs of projections of $X$ onto a top $k$ and a bottom $d - k$ eigenvector of $\Sigma$, (ii) the eigenvalue gaps between a top $k$ and a bottom $d - k$ eigenvalue of $\Sigma$. The second term is the product of the largest eigenvalue of $\mathrm{Cov}(H)$ and $\log(1/\delta)$. In typical regimes where $\delta$ is moderately small, this second term is much smaller than the first. Returning to the question we raised in the abstract, Corollary 1 thus identifies the first term as the key property of the distribution of $X$ that determines the excess risk of PCA, at least in the large sample regime.

Our goal in the next sections will be to derive a non-asymptotic upper bound on the $1 - \delta$ quantile of the excess risk that matches its expression from Corollary 1. We conclude this section by offering two remarks highlighting other aspects of Theorem 1, accompanied by an example illustrating our result on the spiked covariance model.

*Remark* 1 (Empirical projectors). In some applications it is of more interest to measure the performance of PCA through the closeness of the empirical projector $U_n U_n^T$ to the population one $U_* U_*^T$ in a given norm. Dauxois et al. [DPR82] derive the exact asymptotic distribution of $\sqrt{n}(U_n U_n^T - U_* U_*^T)$ from which the result we are about to discuss can potentially be deduced. Here we would like to point out that the asymptotic normality result of Theorem 1 can also be used to establish that as $n \to \infty$,

$$\sqrt{n} \cdot \|U_n U_n - U_* U_*^T\|_p \xrightarrow{d} 2^{1/p} \|G\|_p,$$

for all $p \in [1, \infty]$, where $G$ is the Gaussian matrix defined in Theorem 1 and $\|G\|_p$ is the Schatten-$p$ norm of $G$, i.e. the $p$-norm of its singular values. Compare with equation (2.22) in [RW20].

The case $p = 2$ corresponds to the Frobenius norm $\|U_n U_n - U_* U_*^T\|_F$, and a statement analogous to Corollary 1 holds. Specifically, for $\delta \in [0, 0.1)$ it holds that

$$\lim_{n \to \infty} n \cdot Q_{\mathcal{P}_n}(1 - \delta) \asymp 4 \sum_{i=1}^{d-k} \sum_{j=1}^{k} \frac{\mathrm{E}[\langle u_{k+i}, X \rangle^2 \langle u_j, X \rangle^2]}{(\lambda_j - \lambda_{k+i})^2} + 8\tau^2 \log(1/\delta),$$

where $\mathcal{P}_n := \|U_n U_n - U_* U_*^T\|_F^2$, $\tau^2 = \sup_{\|A\|_F=1} \mathrm{E}[\langle A, G \rangle_F^2]$, and the constants are the same as in Corollary 1. A similar result can be obtained for the other values of $p$ using the noncommutative Khintchine inequality [Tro+15; Van17], though the upper and lower bounds differ by a logarithmic factor in the dimension $d$ for large values of $p$.

**Example 1** (Spiked covariance model). As an application of Theorem 1, we consider the spiked covariance model [Joh01; Nad08]. Specifically, we assume that $X = Z + \varepsilon$ such that $Z$ and $\varepsilon$ are independent, $\varepsilon \sim \mathcal{N}(0, \sigma^2 I_d)$, and $S = \mathrm{E}[ZZ^T]$ has rank $k$. Then $\Sigma = S + I$, the support of $Z$ is contained in the $k$-dimensional subspace spanned by $(u_j)_{j=1}^k$, and $\lambda_j = \eta_j + \sigma^2$ where $(\eta_j)$ are the eigenvalues of $S$ ordered non-increasingly. Taking $\xi_j := \langle Z, u_j \rangle$ we recover the standard form

$$X = \sum_{j=1}^{k} \xi_j u_j + \varepsilon.$$

This spiked covariance model captures the scenario where we observe a noisy version $X$ of the true lower dimensional data point $Z$ corrupted with isotropic noise $\varepsilon$. The goal is to recover, using PCA, the subspace $\mathrm{span}(\{u_j \mid j \in [k]\})$ on which the noise-free data $Z$ is supported. In this setting, Theorem 1 simplifies significantly. Specifically, under this model, the Gaussian matrices $G$ and $H$ in the statements (9) and (10) have independent entries with variances

$$\mathrm{E}[G_{ij}^2] = \sigma^2(1 + \sigma^2/\eta_j^2), \qquad \mathrm{E}[H_{ij}^2] = \sigma^2(\eta_j + \sigma^2/\eta_j),$$

which are constant along rows. From Remark 1 and Theorem 1, we have the distributional results

$$\sqrt{n} \cdot \|U_n U_n^T - U_* U_*^T\|_p \xrightarrow{d} 2^{1/p} \|G\|_p, \qquad n \cdot [R([U_n]) - R([U_*])] \xrightarrow{d} \frac{1}{2} \|H\|_F^2,$$

as $n \to \infty$. The asymptotic quantiles of $\mathcal{P}_n = \|U_n U_n^T - U_* U_*^T\|_F^2$ have the equivalent expression

$$\lim_{n \to \infty} n \cdot Q_{\mathcal{P}_n}(1 - \delta) \asymp 4\sigma^2(d - k) \sum_{j=1}^{k} (1 + \sigma^2/\eta_j^2) + 8\sigma^2(1 + \sigma^2/\eta_k^2) \log(1/\delta), \qquad (11)$$

while those of the excess risk $\mathcal{E}_n = R([U_n]) - R([U_*])$ have the equivalent expression

$$\lim_{n \to \infty} n \cdot Q_{\mathcal{E}_n}(1 - \delta) \asymp \sigma^2(d - k) \sum_{j=1}^{k} (\eta_j + \sigma^2/\eta_j) + \sigma^2 \max_{j \in [k]}(\eta_j + \sigma^2/\eta_j) \log(1/\delta),$$

both for $\delta \in [0, 0.1)$ and the same constants as in Corollary 1. We conclude this example by noting that, using a recent result of Latała et al. [LHY18] and leveraging the independence of the entries of $G$, an analogue of (11) can be derived for large values of $p$ without suffering from the inefficiency of the noncommutative Khintchine inequality highlighted at the end of Remark 1.

*Remark* 2 (Generalized PCA). While our results are framed for PCA, we remark here that they apply to the more general problem of estimating the leading $k$-dimensional eigenspace of a symmetric matrix. Specifically, let $(A_i)_{i=1}^n$ be i.i.d. realizations of a random symmetric matrix $A$, and suppose that we are interested in estimating the leading $k$-dimensional eigenspace of $M := \mathrm{E}[A]$. A natural and common procedure is to estimate it using the leading $k$-dimensional eigenspace of $M_n := n^{-1} \sum_{i=1}^n A_i$. While the reconstruction loss does not make sense for this problem, we may still cast this procedure as an instance of ERM where the loss is given by the negative block Rayleigh quotient. The population and empirical risks are then given by

$$F([U]) = -\frac{1}{2} \mathrm{Tr}(U^T M U), \quad F_n([U]) = -\frac{1}{2} \mathrm{Tr}(U^T M_n U). \qquad (12)$$

PCA then corresponds to the special case $A = XX^T$ where $X$ is a random vector, and the population and empirical reconstruction risks are, up to additive constants, equal to those in (12). Theorem 1 applies almost verbatim to this generic setting, with the only change being that (8) is generalized to

$$\Lambda_{ijst} = \mathrm{E}[(u_{k+i}^T A u_j) \cdot (u_{k+s}^T A u_t)]. \qquad (13)$$

As an example different from PCA, consider the case where $M$ is the adjacency matrix of an undirected weighted graph with non-negative weights. Suppose that we observe $n$ i.i.d. edges $\{J_i, K_i\}_{i=1}^n$ of the graph, sampled from the distribution on the edges that is proportional to their weights. Then one may take $A_i = e_{J_i} e_{K_i}^T + e_{K_i} e_{J_i}^T$, and Theorem 1 with (8) replaced by (13) applies. A similar argument can be made for the estimation of the trailing $k$-dimensional eigenspace of the Laplacian matrix. As examples of potential applications, we mention spectral clustering [e.g. NJW01], community detection [e.g. Abb18], and contrastive learning [e.g. Hao+21].

# 4 Self-concordance of the block Rayleigh quotient

The main ingredient behind Theorem 1 is a Taylor expansion of the population and excess risks at $[U_n]$ around $[U_*]$, which becomes exact in the large sample limit. In order to make Corollary 1 non-asymptotic, an explicit control of the error in these expansions in a reasonably large neighbourhood of $[U_*]$ is what is needed. In this section, we show that the population and empirical reconstruction risks are geodesically generalized self-concordant, in a sense analogous to the one introduced by Bach [Bac10]. This provides the needed control for the non-asymptotic analysis. As this self-concordance result can potentially be of broader interest, we frame it here in more general terms.

Let $A$ be a $d \times d$ symmetric matrix, and let $(v_j)$ be a basis of eigenvectors of $A$ ordered non-increasingly in terms of their eigenvalues $(\mu_j)$. Recall that eigenvectors corresponding to the largest eigenvalue are maximizers of the Rayleigh quotient. The following known construction is a generalization of this familiar identity. Let $F : \mathrm{Gr}(d, k) \to \mathbb{R}$ be given by

$$F([V]) = -\operatorname{Tr}(V^T A V)/2. \tag{14}$$

To see how $F$ relates to the reconstruction risk, note that it can be expressed in terms of it

$$R([U]) = \mathrm{E}\big[\|X - UU^T X\|_2^2\big]/2 = \mathrm{E}[\|X\|_2^2]/2 - \operatorname{Tr}(U^T \Sigma U)/2.$$

The trace expression in (14) is known as the block Rayleigh quotient of $A$. Let $k_1$ and $k_2$ be the smallest and largest indices $i$ such that $\mu_i = \mu_k$ respectively. The set of minimizers of $F$ is given by (see for example [Tao12, Proposition 1.3.4])

$$\mathscr{V}_* := \{[V] \in \mathrm{Gr}(d, k) \mid \mathrm{col}(V) = \mathrm{span}(v_1, \dots, v_{k_1-1}) \oplus S, S \subset \mathrm{span}(v_{k_1}, \dots, v_{k_2})\} \tag{15}$$

where $\oplus$ is the direct sum of subspaces, and $S$ is a subspace of dimension $k - k_1 + 1$. In the case where $\mu_k > \mu_{k+1}$ that we have been operating under, $k_1 = k_2 = k$ and $\mathscr{V}_*$ becomes a singleton.

Recall the definition of geodesics and principal angles from Section 2.1. The following is the main result of this section [c.f. Bac10, Lemma 1].

**Proposition 1** (Generalized self-concordance of the block Rayleigh quotient). *Assume* $\mu_k > \mu_{k+1}$, *let* $[V_*]$ *be the global minimizer of* $F$, *and let* $[V] \in \mathrm{Gr}(d, k) \setminus [V_*]$ *such that* $\theta_k([V_*], [V]) < \pi/4$. *Let* $g(t) := F(\gamma(t))$ *where* $\gamma(t)$ *is either* $\mathrm{Exp}_{[V_*]}(t \mathrm{Log}_{[V_*]}([V]))$ *or* $\mathrm{Exp}_{[V]}(t \mathrm{Log}_{[V]}([V_*]))$. *Then*

$$|g'''(t)| \le 2\theta \cdot \tan(2t\theta) \cdot g''(t),$$

*for all* $t \in [0, 1]$ *where we shortened* $\theta_k([V_*], [V])$ *to* $\theta$. *As a consequence,*

$$\frac{\sin^2(\theta)}{\theta^2} \cdot \frac{g''(0)}{2} \le g(1) - g(0) - g'(0) \le \psi(\theta) \cdot \frac{g''(0)}{2},$$

*where*

$$\psi(\theta) := \theta^{-1} \int_0^1 \log[\tan(\theta t + \pi/4)]\, \mathrm{d}t,$$

*satisfies* $\psi(\theta) \to 1$ *as* $\theta \to 0$ *and* $\psi(\theta) \to c$ *as* $\theta \to \pi/4$ *for* $c \approx 1.485$.

In words, Proposition 1 says that for any $[V] \in \mathrm{Gr}(d, k)$ that is less than $\pi/4$ away from the minimizer of $F$ in maximum principal angle, the restriction of $F$ to the geodesic connecting $[V]$ to this minimizer is well approximated by its second order Taylor expansion, up to a factor of approximately 2 in the second term of this expansion. We have the immediate corollary [cf. Bac10, Proposition 1].

**Corollary 2.** *In the setting of Proposition 1, the following estimates hold.*

$$F([V]) \ge F([V_*]) + \langle \mathrm{grad}\, F([V_*]), \xi \rangle_{[V_*]} + \frac{4}{5} \cdot \frac{1}{2} \langle \mathrm{Hess}\, F([V_*])[\xi], \xi \rangle_{[V_*]},$$

$$F([V]) \le F([V_*]) + \langle \mathrm{grad}\, F([V_*]), \xi \rangle_{[V_*]} + \frac{3}{2} \cdot \frac{1}{2} \langle \mathrm{Hess}\, F([V_*])[\xi], \xi \rangle_{[V_*]},$$

*where* $\xi = \mathrm{Log}_{[V_*]}([V])$, *and where for any* $[U] \in \mathrm{Gr}(d, k)$ *and* $\zeta \in T_{[U]} \mathrm{Gr}(d, k)$ *with* $\Delta = \mathrm{lift}_U(\zeta)$

$$\mathrm{lift}_U(\mathrm{grad}\, F([U])) = -(I - UU^T)AU, \quad \mathrm{lift}_U(\mathrm{Hess}\, F([U])[\zeta]) = \Delta U^T A U - (I - UU^T)A\Delta.$$

*The statement remains true when* $[V]$ *and* $[V_*]$ *are interchanged in the above inequalities.*

Related results include those of [ZJS16] who showed that $F$ satisfies a version of the Polyak–Łojasiewicz inequality for $k = 1$, and [AV24] who showed that $F$, when restricted as in Proposition 1, satisfies a version of strong convexity - Proposition 1 was inspired by the latter work. The result of the next section builds on Corollary 2 to provide a non-asymptotic analogue of Corollary 1.

# 5 Non-asymptotic bound

The following is the second main result of the paper. The parameters $\mathcal{V}$ and $\nu$ appearing in it are defined in Remark 3 below. We compute them explicitly under a Gaussian model in Example 2.

**Theorem 2.** *Assume that* $\lambda_{k+1} > \lambda_k$, $\mathrm{E}[X_j^4] < \infty$ *for all* $j \in [d]$, *and let* $\delta \in [0,1)$. *If*

$$n \geq (32\mathcal{V} + 4)\log(3k(d-k)) + (16\nu + 8)\log(4/\delta) + \frac{16(\mathcal{S} + r(n))}{\delta(\lambda_k - \lambda_{k+1})^2}, \qquad (16)$$

*then with probability at least* $1 - \delta$

$$R([U_n]) - R([U_*]) \leq \frac{75}{n \cdot \delta} \sum_{i=1}^{d-k} \sum_{j=1}^{k} \frac{\mathrm{E}[\langle u_{k+i}, X \rangle^2 \langle u_j, X \rangle^2]}{\lambda_j - \lambda_{k+i}}, \qquad (17)$$

*where for* $c(d) = 4(1 + 2\lceil \log(d) \rceil)$,
$$\mathcal{S} := c(d) \cdot \|\mathrm{E}[(XX^T - \Sigma)^2]\|_{\mathrm{op}}, \quad r(n) := c^2(d) \cdot n^{-1} \, \mathrm{E}[\max_{i \in [n]} \|X_i X_i^T - \Sigma\|_{\mathrm{op}}^2].$$

Focusing on (17), Theorem 2 says that, up to a worse dependence on $\delta$, the tight asymptotic upper bound on the $1 - \delta$ quantile of the excess risk in Corollary 1 holds true for finitely many samples, provided that the sample size is larger than a certain distribution-dependent constant. Given the weak moment condition assumed, the dependence on $\delta$ in Theorem 2 is likely unimprovable. It is possible to obtain a $\log(1/\delta)$ dependence as in Corollary 1 under the assumption that $X$ is bounded. We favour the above statement as it highlights an important shortcoming of ERM, and thus PCA: its performance degrades under heavy-tailed data - we refer the interested reader to the literature on robust estimation [e.g. LM19]. The results in [RW20] are the most closely related, though they do not capture the right dependence on the distribution of $X$ identified in Corollary 1 and recovered in (17). They however hold under different assumptions and can cover a wider range of sample sizes.

The sample size restriction (16) consists of three terms. They arise from two distinct steps of the analysis: a global and a local one. The global one ensures that with high probability, $[U_n]$ is within a maximum principal angle of $\pi/4$ from $[U_*]$. This step is carried out using standard existing tools - namely the Davis-Kahan theorem [e.g. YWS15] - and is likely loose. It results in the third term of (16), the largest of the three. The second step is a local analysis that uses our new self-concordance result from Proposition 1, and is where our original contribution lies. This step results in the first two terms of (16), the first of which typically dominates: their role is to ensure that the curvature of the empirical risk at $[U_*]$ is strong enough to force $[U_n]$ to be near it. Qualitatively, the explicit expression of $\mathcal{V}$ and $\nu$ in Remark 3 below indicate that they induce a quadratic dependence on the inverse of the eigengap on the sample size restriction. Example 2 below gives an easily interpretable expression for $\mathcal{V}$ and $\nu$ in the special case when $X$ is Gaussian and centered.

*Remark* 3 (Variance parameters). The parameters $\mathcal{V}$ and $\nu$ appearing in Theorem 2 admit an explicit expression, though it is quite involved in the general case. Recall from Theorem 1 the definition of the eigengaps $\delta_{ij} = \lambda_j - \lambda_{k+i}$. Let $\widetilde{X}$ denote the coordinates of $X$ in the basis of eigenvectors $(u_j)$, i.e. $\widetilde{X}_j := \langle X, u_j \rangle$. Define and recall from Theorem 1

$$\Gamma_{jsrp} := \mathrm{E}[\widetilde{X}_j \widetilde{X}_s \widetilde{X}_r \widetilde{X}_p], \quad \Lambda_{ijts} = \mathrm{E}[\widetilde{X}_{k+i} \widetilde{X}_j \widetilde{X}_{k+t} \widetilde{X}_s], \quad \Omega_{itql} := \mathrm{E}[\widetilde{X}_{k+i} \widetilde{X}_{k+t} \widetilde{X}_{k+q} \widetilde{X}_{k+l}],$$

for $j, s, r, p \in [k]$ and $i, t, q, l \in [d-k]$. These form a subset of the fourth order moments of $\widetilde{X}$. Let

$$\mathcal{V} := \sup_{\|M\|_F = 1} \sum_{j,r,p} a_{jrp}(M) \cdot \Gamma_{jjrp} - 2 \sum_{i,j,t,s} b_{ijts}(M) \cdot \Lambda_{ijts} + \sum_{i,q,l} c_{iql}(M) \cdot \Omega_{iiql}, \qquad (18)$$

where the coefficients are given by
$$a_{jrp}(M) := \sum_i \frac{m_{ir} m_{ip}}{\delta_{i,j}\sqrt{\delta_{ir}\delta_{ip}}}, \quad b_{ijts}(M) := \sum \frac{m_{is} m_{tj}}{\delta_{ij}\sqrt{\delta_{is}\delta_{tj}}}, \quad c_{iql}(M) := \sum_j \frac{m_{qj} m_{lj}}{\delta_{ij}\sqrt{\delta_{qj}\delta_{lj}}}.$$

Finally, define the parameter
$$\nu := \sup_{\|M\|_F = 1} \sum_{j,s,r,p} \alpha_{jsrp}(M) \cdot \Gamma_{jsrp} - 2 \sum_{t,s,q,r} \beta_{tsqr}(M) \cdot \Lambda_{tsqr} + \sum_{i,t,q,l} \kappa_{itql}(M) \cdot \Omega_{itql}, \qquad (19)$$

where the coefficients are given by, suppressing the dependence on $M$
$$\alpha_{jsrp} := \sum_{i,t} \frac{m_{ij} m_{is} m_{tr} m_{tp}}{\sqrt{\delta_{ij}\delta_{is}\delta_{tr}\delta_{tp}}}, \beta_{tsqr} := \sum_{i,j} \frac{m_{tj} m_{is} m_{qj} m_{ir}}{\sqrt{\delta_{tj}\delta_{is}\delta_{qj}\delta_{ir}}}, \kappa_{itql} := \sum_{j,s} \frac{m_{ij} m_{tj} m_{qs} m_{ls}}{\sqrt{\delta_{ij}\delta_{tj}\delta_{qs}\delta_{ls}}}.$$

Typically, $\mathcal{V}$ is much larger than $\nu$, and we always have $\mathcal{V} \geq \nu$.

**Example 2** (Gaussian model). The variance parameters appearing in Theorem 2 and defined in Remark 3 simplify greatly when $X$ has mean zero and its coordinates in a basis of eigenvectors of $\Sigma$ are independent. When $X \sim \mathcal{N}(0, \Sigma)$ we can compute them exactly in terms of the spectrum of $\Sigma$:

$$\mathcal{V} = \sum_{s=1}^{k} \frac{(1 + \mathbb{I}[s = k])\lambda_k \lambda_s}{(\lambda_k - \lambda_{k+1})(\lambda_s - \lambda_{k+1})} + \sum_{t=1}^{d-k} \frac{(1 + \mathbb{I}[t = 1])\lambda_{k+1}\lambda_{k+t}}{(\lambda_k - \lambda_{k+1})(\lambda_k - \lambda_{k+t})}, \tag{20}$$

$$\nu = \max_{i \in [d-k]} \max_{j \in [k]} \frac{\lambda_j^2 + \lambda_{k+i}^2}{(\lambda_j - \lambda_{k+i})^2}. \tag{21}$$

## 6 Discussion

We started this paper with a simple question: which property of the distribution of $X$ governs the performance of PCA as measured by its excess risk? In the large sample limit, and under very mild assumptions, we found an equally simple answer (see (10) and Corollary 1):

$$\sum_{i=1}^{d-k} \sum_{j=1}^{k} \frac{\mathrm{E}[\langle u_{k+i}, X \rangle^2 \langle u_j, X \rangle^2]}{\lambda_j - \lambda_{k+i}}. \tag{22}$$

Our second main contribution was to derive an upper bound on the critical sample size - that beyond which the excess risk of PCA is governed by (22) (see (16)). In the general case, this bound admits an explicit expression in terms of the fourth moments of $X$ and the eigengaps appearing in (22) (see Remark 3), though its precise description is quite involved. In the special case of Gaussian $X$, we showed that the terms in this bound take on an exceptionally simple form (see Example 2).

There are two main limitations of our results. The first is that they rely on the eigengap condition $\lambda_{k+1} - \lambda_k > 0$. While this is a mild assumption, it would be desirable to relax it, though this is quite challenging with our approach. To see why, note that without it, the minimizers of the reconstruction risk form a submanifold (itself a Grassmannian) of $\mathrm{Gr}(d, k)$ (see (15)). The classical theory of asymptotic statistics, upon which our results rely, does not immediately apply in such a degenerate setting [Van+96; Van00], and we leave this problem to future work.

The second limitation we would like to point out is related to Theorem 2. As described after its statement, the global component of the analysis leading to it is unlikely to be tight. To accurately capture the sample complexity of this step, we suspect that one would need to leverage analytical properties of the reconstruction risk as we do in our local analysis. This is however quite challenging as globally the reconstruction risk is ill-behaved and has for example many critical points [SI14]. We anticipate that new insights are needed to fully capture the sample complexity of this global step.

Finally, let us mention that while the setting we consider is both classical and quite general, there are potentially interesting cases that our framework does not cover. For example, our approach does not directly apply to Kernel PCA [SSM98] or functional PCA [RS02], and extending it to these settings would require us to work with an infinite-dimensional analogue of the Grassmannian - a daunting task. Similarly our focus is on characterizing the performance of PCA on a fixed but unknown data distribution, and to do this in as much generality as possible. This in contrast with the literature on high-dimensional PCA which typically considers sequences of Gaussian problems indexed by their dimension, but provides potentially finer-grained results [e.g. Joh01; Pau07; CMW13]. Finally, we mention that in practice, PCA is typically performed with an initial centering step. This neatly fits in our setting by changing the search space from the Grassmannian to the Grassmannian of affine subspaces [LWY21], though some work is required to make this approach viable.

Beyond addressing the limitations discussed above, there are a few directions that are potentially worth exploring. From an analysis standpoint, the output of PCA - along with its generalized version described in Remark 2 - is often used as a preprocessing step for downstream tasks such as regression or classification. It would be interesting to investigate whether the techniques developed in this paper can be extended to provide end-to-end guarantees for such two-stage procedures. Another intriguing direction would be to explore whether the self-concordance result established in Proposition 1 can be leveraged to obtain improved convergence guarantees for optimization algorithms applied to the block Rayleigh quotient (14), particularly for Edelman et al. [EAS98]'s version of Newton's method.

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

## A  Further background on the Grassmannian

In Section 2, we focused on the properties of the Grassmannian necessary to state our results. For the analysis we require more tools, which we briefly describe here. As before, we adopt a computation-oriented description, and refer the reader to Chapter 10 in [Bou23] for a rigorous treatment.

**Parallel transport.**  Let $[U] \in \mathrm{Gr}(d, k)$ and $\xi \in T_{[U]} \mathrm{Gr}(d, k)$, and consider the curve,

$$\alpha(t) = UQ \cos(tS)Q^T + P \sin(tS)Q^T,$$

for $t \in [0, 1]$, and where $\mathrm{lift}_U(\xi) = PSQ^T$ is a SVD. This is the geodesic starting at $[U]$ in direction $\xi$ as defined in (6) at the level of representatives in $\mathrm{St}(d, k)$, i.e. $[\alpha(t)] = \mathrm{Exp}_{[U]}(t\xi)$. For each $t \in [0, 1]$ and $\zeta \in T_{[U]} \mathrm{Gr}(d, k)$, define the map $P_{\xi,t}$ by

$$\mathrm{lift}_{\alpha(t)}(P_{\xi,t}(\zeta)) := (-UQ \sin(tS)P^T + P \cos(tS)P^T + I_d - PP^T) \mathrm{lift}_U(\zeta). \tag{23}$$

This is the parallel transport map along the geodesic $[\alpha(t)]$. See equation (3.18) in [BZA24]. For any fixed $t$, $\zeta \mapsto P_{\xi,t}(\zeta)$ is an invertible linear map that preserves the inner product between $T_{[U]} \mathrm{Gr}(d, k)$ and $T_{[\alpha(t)]} \mathrm{Gr}(d, k)$. Informally, $t \mapsto P_{\xi,t}(\zeta)$ transports $\zeta$ along the tangent spaces of $[\alpha(t)]$ such that it stays "constant", i.e. the derivative of $t \mapsto P_{\xi,t}(\zeta)$, properly defined, is zero.

For the rest of this section, let $\tilde{f} : \mathbb{R}^{d \times k} \to \mathbb{R}$ be such that $\tilde{f}(U) = \tilde{f}(UQ)$ for any orthogonal $Q \in \mathbb{R}^{k \times k}$ and $U \in \mathrm{St}(d, k)$. We define $f : \mathrm{Gr}(d, k) \to \mathbb{R}$ by $f([U]) := \tilde{f}(U)$. We assume that $f$ is sufficiently smooth, under the proper notion of smoothness, to justify the computations below.

**Gradients and Hessians.**  The Riemannian gradient of $f$ at $[U]$ is the element of $T_{[U]} \mathrm{Gr}(d, k)$ given by

$$\mathrm{lift}_U(\mathrm{grad}\, f([U])) = (I_d - UU^T)\nabla \tilde{f}(U), \tag{24}$$

where $\nabla \tilde{f}$ is the Euclidean gradient of $\tilde{f}$. See equation (9.84) in [Bou23]. Similarly, the Riemannian Hessian of $f$ is the linear map $\mathrm{Hess}\, f([U]) : T_{[U]} \mathrm{Gr}(d, k) \to T_{[U]} \mathrm{Gr}(d, k)$ given by

$$\mathrm{lift}_U(\mathrm{Hess}\, f([U])[\xi]) = (I_d - UU^T)\nabla^2 \tilde{f}(U)[\mathrm{lift}_U(\xi)] - \mathrm{lift}_U(\xi)U^T \nabla \tilde{f}(U), \tag{25}$$

for any $\xi \in T_{[U]} \mathrm{Gr}(d, k)$, and where $\nabla^2 \tilde{f}$ is the Euclidean Hessian of $\tilde{f}$, viewed as a linear map $\mathbb{R}^{d \times k} \to \mathbb{R}^{d \times k}$. See equation (9.86) in [Bou23].

**Higher order derivatives.**  The total $s$-th order covariant derivative of $f$ is denoted by $\nabla^m f$. See Definition 10.77 and Example 10.78 in [Bou23]. It is a map that takes an $m+1$ tuple $([U], \xi_1, \ldots, \xi_m)$ where $[U] \in \mathrm{Gr}(d, k)$ and $\xi_j \in T_{[U]} \mathrm{Gr}(d, k)$ for all $j \in [m]$ and outputs a real number. It is linear in each of its last $m$ arguments, and can be computed iteratively as follows: $\nabla^0 f = f$ and

$$\nabla^m f([U])(\xi_1, \ldots, \xi_m) = \frac{\mathrm{d}}{\mathrm{d}t}\nabla^{m-1}f(\mathrm{Exp}_{[U]}(t\xi_m), P_{\xi_m,t}(\xi_1), \ldots, P_{\xi_m,t}(\xi_{m-1}))\Big|_{t=0}. \tag{26}$$

See equation (10.53) in [Bou23]. In particular, for $m = 1$ and $m = 2$ we have the identities

$$\nabla f([U], \xi) = \langle \mathrm{grad}\, f([U]), \xi \rangle_{[U]}, \quad \nabla^2 f([U], \xi_1, \xi_2) = \langle \mathrm{Hess}\, f([U])[\xi_1], \xi_2 \rangle_{[U]}. \tag{27}$$

See Example 10.78 in [Bou23].

**Taylor expansions.**  Let $[U] \in \mathrm{Gr}(d, k)$ and $\xi \in T_{[U]} \mathrm{Gr}(d, k)$, and consider the geodesic $\gamma(t) = \mathrm{Exp}_{[U]}(t\xi)$ for $t \in [0, 1]$. Define the function $g : [0, 1] \to \mathbb{R}$ by $g(t) := f(\gamma(t))$. Then we have

$$g'(t) = \nabla f(\gamma(t), P_\xi(\xi)), \quad g''(t) = \nabla^2 f(\gamma(t), P_{\xi,t}(\xi), P_{\xi,t}(\xi)), \tag{28}$$
$$g'''(t) = \nabla^3 f(\gamma(t), P_{\xi,t}(\xi), P_{\xi,t}(\xi), P_{\xi,t}(\xi)).$$

See Example 10.81 in [Bou23]. By Taylor's theorem applied to $g$ around $0$ and (27) we have

$$f(\mathrm{Exp}_{[U]}(\xi)) = f([U]) + \langle \mathrm{grad}\, f([U]), \xi \rangle_{[U]} + \frac{1}{2}\langle \mathrm{Hess}\, f([U])[\xi], \xi \rangle_{[U]}$$
$$+ \frac{s^3}{6}\nabla^3 f(\gamma(s), P_{\xi,s}(\xi), P_{\xi,s}(\xi), P_{\xi,s}(\xi)), \tag{29}$$

for some $s \in [0, 1]$, and where we used the mean value form of the remainder. We also have the following Taylor expansion the gradient of $f$

$$P_{\xi,1}^{-1}(\operatorname{grad} f(\operatorname{Exp}_{[U]}(\xi))) = \operatorname{grad} f([U]) + \operatorname{Hess} f([U])[\xi]$$
$$+ \int_0^1 \{P_{\xi,s}^{-1} \circ \operatorname{Hess} f(\gamma(s)) \circ P_{\xi,s} - \operatorname{Hess} f([U])\}[\xi] \, \mathrm{d}s. \quad (30)$$

See Step 2 in the proof of Proposition 10.55 in [Bou23].

**Lipschitz continuous derivatives.** The function $f$ is said to have an $L$-Lipschitz continuous $m$-th derivative if

$$\sup_{[U] \in \operatorname{Gr}(d,k)} \sup_{\|\xi_j\|=1, j \in [m+1]} |\nabla^{m+1} f([U], \xi_1, \ldots, \xi_m)| \leq L. \quad (31)$$

See Proposition 10.83 in [Bou23]. For the special case $m = 2$, this is equivalent to Hessian $L$-Lipschitzness, which states that for all $[U] \in \operatorname{Gr}(d, k)$ and $\xi \in T_{[U]} \operatorname{Gr}(d, k)$

$$\|P_{\xi,1}^{-1} \circ \operatorname{Hess} f(\operatorname{Exp}_{[U]}(\xi)) \circ P_{\xi,1} - \operatorname{Hess} f([U])\|_{\mathrm{op}} \leq L\|\xi\|.$$

See Exercise 10.89 in [Bou23].

# B   Analysis of the block Rayleigh quotient

In this section, we state and prove the two main technical results behind Theorems 1 and 2. We state them here in terms of the negative block Rayleigh quotient (14). We will use them in subsequent sections on the empirical and population reconstruction risk, which we recall from Section 4 are up to an additive constant equal to the negative block Rayleigh quotient of $\Sigma_n$ and $\Sigma$ respectively.

Recall the setup of Section 4. We have a symmetric matrix $A \in \mathbb{R}^{d \times d}$ and its associated negative block Rayleigh quotient $F : \operatorname{Gr}(d, k) \to \mathbb{R}$ given by

$$F([V]) = -(1/2) \operatorname{Tr}(V^T A V).$$

In the context of Appendix A, this function can be obtained from the one defined on Euclidean space $\tilde{F} : \mathbb{R}^{d \times k} \to \mathbb{R}$ given by $\tilde{F}(B) = -(1/2) \operatorname{Tr}(B^T A B)$. Hence the Riemannian gradient and Hessian of $F$ are given by, using (24) and (25),

$$\operatorname{lift}_V(\operatorname{grad} F([V])) = -(I_d - VV^T)AV \quad (32)$$
$$\operatorname{lift}_V(\operatorname{Hess} F([V])[\xi]) = \Delta V^T A V - (I_d - VV^T)A\Delta \quad (33)$$

where $\Delta = \operatorname{lift}_V(\xi)$.

## B.1   Hessian Lipschitzness of the block Rayleigh quotient

Recall the discussion on the higher order derivatives $\nabla^s F$ of $F$ from Appendix A.

**Proposition 2.** *For all $[V] \in \operatorname{Gr}(d, k)$ and $\xi_1, \xi_2, \xi_3 \in T_{[V]} \operatorname{Gr}(d, k)$, it holds that*

$$\nabla^3 F([V], \xi_1, \xi_2, \xi_3) = \langle A, V[\Delta_1^T \Delta_2 \Delta_3^T + \Delta_2^T \Delta_1 \Delta_3^T + \Delta_3^T \Delta_1 \Delta_2^T + \Delta_3^T \Delta_2 \Delta_1^T]\rangle_F,$$

*where $\Delta_j = \operatorname{lift}_V(\xi_j)$ for $j \in \{1, 2, 3\}$. As a consequence*

$$\sup_{[V] \in \operatorname{Gr}(d,k)} \sup_{\|\xi_1\|=1, \|\xi_2\|=1, \|\xi_3\|=1} |\nabla^3 F([V], \xi_1, \xi_2, \xi_3)| \leq 4\|A\|_F,$$

*and for all $[V] \in \operatorname{Gr}(d, k)$ and $\xi \in T_{[V]} \operatorname{Gr}(d, k)$,*

$$\|P_{\xi,1}^{-1} \circ \operatorname{Hess} F(\operatorname{Exp}_{[V]}(\xi)) \circ P_{\xi,1} - \operatorname{Hess} F([V])\|_{\mathrm{op}} \leq 4\|A\|_F\|\xi\|.$$

*Proof.* Fix $[V] \in \operatorname{Gr}(d, k)$ and $\xi_1, \xi_2, \xi_3 \in T_{[V]} \operatorname{Gr}(d, k)$. Then we have by (26)

$$\nabla^3 F([V], \xi_1, \xi_2, \xi_3) = \frac{\mathrm{d}}{\mathrm{d}t} \nabla^2 F(\operatorname{Exp}_{[V]}(t\xi_3), P_{\xi_3,t}(\xi_1), P_{\xi_3,t}(\xi_2))\Big|_{t=0}. \quad (34)$$

Let $\Delta_3 := \mathrm{lift}_V(\xi_3) = PSQ^T$ be a SVD, $\Delta_1 := \mathrm{lift}_V(\xi_1)$, and $\Delta_2 := \mathrm{lift}_V(\xi_2)$. Define

$$V(t) := VQ\cos(tS)Q^T + P\sin(tS)Q^T, \quad B(t) := -VQ\sin(tS)P^T + P\cos(tS)P^T - PP^T,$$
$$\Delta_1(t) := \mathrm{lift}_{V(t)}(P_{\xi_3,t}(\xi_1)) = B(t)\Delta_1 + \Delta_1, \quad \Delta_2(t) := \mathrm{lift}_{V(t)}(P_{\xi_3,t}(\xi_2)) = B(t)\Delta_2 + \Delta_2,$$

where $[V(t)] = \mathrm{Exp}_{[V]}(t\xi_3)$ by (6) and where we used (23) for the parallel transport map. Using these definitions, (27), (33), and the fact that the map $\mathrm{lift}$ preserves the inner product

$$\nabla^2 F(\mathrm{Exp}_{[U]}(t\xi_3), P_{\xi_3,t}(\xi_1), P_{\xi_3,t}(\xi_2)) = \langle \Delta_1(t)V(t)^T AV(t) - A\Delta_1(t), \Delta_2(t)\rangle_F, \qquad (35)$$

where we used the identity $(I_d - V(t)V(t)^T)\Delta_2(t) = \Delta_2(t)$, which holds since $V(t)^T\Delta_2(t) = 0$ by (4), to simplify the resulting expression. Now taking derivatives with respect to $t$, dropping the dependence on $t$ in the notation, and writing $\dot{V}$ for the derivative of $V(t)$, we get

$$\frac{\mathrm{d}}{\mathrm{d}t}\langle \Delta_1 V^T AV - A\Delta_1, \Delta_2\rangle_F = \mathrm{Tr}(\dot{V}^T AV\Delta_1^T\Delta_2) + \mathrm{Tr}(V^T A\dot{V}\Delta_1^T\Delta_2) + \mathrm{Tr}(V^T AV\dot{\Delta}_1^T\Delta_2)$$
$$+ \mathrm{Tr}(V^T AV\Delta_1^T\dot{\Delta}_2) - \mathrm{Tr}(\dot{\Delta}_1^T A\Delta_2) - \mathrm{Tr}(\Delta_1^T A\dot{\Delta}_2). \quad (36)$$

Noting that $\dot{B}(t) = -VQ\cos(tS)SP^T - P\sin(tS)SP^T$, we get

$$\dot{B}(0) = -VQSP^T = -V\Delta_3^T, \quad \dot{\Delta}_1(0) = -V\Delta_3^T\Delta_1, \quad \dot{\Delta}_2(0) = -V\Delta_3^T\Delta_2.$$

Replacing in (36) and simplifying, then using (35) and (34) finishes the proof of the first statement. The second follows from the Cauchy-Schwarz inequality, the inequality $\|VC\|_F \leq \|V\|_{\mathrm{op}}\|C\|_F$ and $\|V\|_{\mathrm{op}} = 1$, the submultiplicativity of the Frobenius norm, and the fact that $\|\mathrm{lift}_V(\xi)\|_F = \|\xi\|$ for all $\xi \in T_{[U]}\mathrm{Gr}(d, k)$. See the end of Appendix A for the last statement. $\qquad \square$

## B.2 Generalized self-concordance of the block Rayleigh quotient

*Proof of Proposition 1.* Recall that $(v_j)_{j=1}^d$ is a basis of eigenvectors of $A$ ordered non-increasingly according to their corresponding eigenvalues $(\mu_j)_{j=1}^d$. Let $V_* \in \mathrm{St}(d, k)$ be the matrix whose $j$-th column is $v_j$. We start with the first statement, and with the case where $\gamma(t) = \mathrm{Exp}_{[V_*]}(t\,\mathrm{Log}_{[V_*]}([V]))$. Define $\xi := \mathrm{Log}_{[V_*]}([V])$, and let $\mathrm{lift}_{V_*}(\xi) = PSQ^T$ be a SVD. Let $r$ be the rank of $\mathrm{lift}_{V_*}(\xi)$, and let $P_r$ be the $d \times k$ matrix whose first $r$ columns match those of $P$, and whose last $k - r$ columns are $0$. Then we have by (6)

$$\gamma(t) = [V_*Q\cos(tS) + P_r\sin(tS)], \qquad (37)$$

where we used that the post-multiplication by $Q^T$, an orthogonal matrix, can be dropped without affecting the equivalence class, and we used that $P\sin(tS) = P_r\sin(tS)$ since the last $k - r$ singular values in $S$ are zero. Now let $V(t) = V_*Q\cos(tS) + P_r\sin(tS)$. Then we have

$$g(t) = F(\gamma(t)) = -\frac{1}{2}\mathrm{Tr}(V(t)^T AV(t)),$$

and its derivatives are given by

$$g'(t) = -\mathrm{Tr}(\dot{V}(t)^T AV(t)),$$
$$g''(t) = -\mathrm{Tr}(\ddot{V}(t)^T AV(t)) - \mathrm{Tr}(\dot{V}(t)^T A\dot{V}(t)),$$
$$g'''(t) = -\mathrm{Tr}(\dddot{V}(t)^T AV(t)) - 3\mathrm{Tr}(\ddot{V}(t)^T A\dot{V}(t)).$$

A straightforward computation shows that

$$\dot{V}(t) = (-V_*Q\sin(tS) + P_r\cos(tS))S, \quad \ddot{V}(t) = -V(t)S^2, \quad \dddot{V}(t) = -\dot{V}(t)S^2.$$

Replacing yields

$$g''(t) = \mathrm{Tr}(S^2 V(t)^T AV(t)) - \mathrm{Tr}(\dot{V}(t)^T A\dot{V}(t)), \quad g'''(t) = 4\mathrm{Tr}(S^2\dot{V}(t)^T AV(t)). \qquad (38)$$

To further simplify this expression, let $V_*^\perp \in \mathrm{St}(d, d - k)$ be the matrix whose $j$-th column is $v_{k+j}$. Then since $V_*^T \mathrm{lift}_{V_*}(\xi) = 0$ by (4) and by definition of $P_r$, we have $V_*^T P_r = 0$. Hence there exists

$\Gamma \in \mathbb{R}^{(d-k)\times k}$ whose first $r$ columns are orthonormal and last $k - r$ columns are zero such that $P_r = V_*^\perp \Gamma$. Finally, we may write an eigendecomposition of $A$ as

$$A = [V_* \mid V_*^\perp] \cdot D \cdot [V_* \mid V_*^\perp]^T,$$

where $D = \mathrm{diag}(\mu_1, \dots, \mu_d)$. Performing block-wise matrix multiplication we obtain the identities

$$V_*^T A V_* = D_{\leq k}, \quad V_*^T A P_r = 0, \quad P_r^T A P_r = \Gamma^T D_{>k} \Gamma. \tag{39}$$

where $D_{<k} = \mathrm{diag}(d_1, \dots, d_k)$ and $D_{>k} = \mathrm{diag}(d_{k+1}, \dots, d_d)$. Replacing in (38) $V(t)$ and $\dot{V}(t)$ by their expressions and using the identities (39) yields

$$g''(t) = \mathrm{Tr}(S^2 \cos(2tS)\{Q^T D_{\leq k} Q - \Gamma^T D_{>k}\Gamma\}), \tag{40}$$

and

$$
\begin{aligned}
|g'''(t)| &= 2|\mathrm{Tr}(S^3 \sin(2tS)\{Q^T D_{\leq k} Q - \Gamma^T D_{>k}\Gamma\})| \\
&= 2|\mathrm{Tr}(S\tan(2tS) \cdot S\sqrt{\cos(2tS)}\{Q^T D_{\leq k} Q - \Gamma^T D_{>k}\Gamma\}S\sqrt{\cos(2tS)})| \\
&= 2|\langle S\tan(2tS), S\sqrt{\cos(2tS)}\{Q^T D_{\leq k} Q - \Gamma^T D_{>k}\Gamma\}S\sqrt{\cos(2tS)}\rangle_F| \\
&\leq 2\|S\tan(2tS)\|_\infty \|S\sqrt{\cos(2tS)}\{Q^T D_{\leq k} Q - \Gamma^T D_{>k}\Gamma\}S\sqrt{\cos(2tS)}\|_1 \\
&= 2\|S\tan(2tS)\|_\infty \cdot g''(t),
\end{aligned}
$$

where in the penultimate line we have used Holder's inequality for Schatten $p$-norms, and in the last we have used the fact that the matrix $S\sqrt{\cos(2tS)}\{Q^T D_{\leq k} Q - \Gamma^T D_{>k}\Gamma\}S\sqrt{\cos(2tS)}$ is positive-semidefinite, so its nuclear norm equals its trace. To see why the latter matrix is positive-semidefinite, note that for any $k$-dimensional unit vector $y$,

$$y^T Q^T D_{\leq k} Q y \geq \mu_k, \quad y^T \Gamma^T D_{>k} \Gamma y \leq \mu_{k+1}$$

where these inequalities follow from the fact that $Qy$ is unit norm, and $\Gamma y$ is at most unit norm by definition of $\Gamma$, and furthermore the singular values in $S$ correspond to the principal angles between $[V_*]$ and $[V]$, which by assumption are less than $\pi/4$, so that $\cos(2tS) > 0$. This last observation also shows that $2\|S\tan(2tS)\|_\infty = 2\theta\tan(2t\theta)$ where $\theta$ is the maximum principal angle between $[V_*]$ and $[V]$. This concludes the proof of the first statement for the case $\gamma(t) = \mathrm{Exp}_{[V_*]}(t\,\mathrm{Log}_{[V_*]}([V]))$.

For the second statement, we first show that $g''(t) > 0$ for all $t \in [0, 1]$. Indeed, expanding the trace expression in (40) we obtain

$$
\begin{aligned}
g''(t) &= \sum_{j=1}^{k} S_j^2 \cos(2tS_j) \cdot \left( \sum_{i=1}^{k} \mu_i Q_{ji}^2 - \sum_{i=1}^{d-k} \mu_{k+i}\Gamma_{ji}^2 \right) \\
&\geq \sum_{j=1}^{k} S_j^2 \cos(2tS_j) \cdot (\mu_k - \mu_{k+1}) \\
&> 0.
\end{aligned}
$$

where in the second line we used that the columns of $Q$ are orthonormal, and that the columns of $\Gamma$ are either of length one or zero, and in the last line we used the assumption $\mu_k - \mu_{k+1} > 0$. We use this result to justify rearranging the first statement of Proposition 1 as follows

$$-2\theta\tan(2t\theta) \leq \frac{\mathrm{d}}{\mathrm{d}t}\log(g''(t)) = \frac{g'''(t)}{g''(t)} \leq 2\theta\tan(2t\theta)$$

Integrating once, exponentiating, then integrating twice yields the second statement in Proposition 1. See the proof of Lemma 1 in [Bac10] for a very similar calculation.

Finally, the case $\gamma(t) = \mathrm{Exp}_{[V]}(t\,\mathrm{Log}_{[V]}([V_*]))$ follows from the first one using the identity $\mathrm{Exp}_{[V]}(t\,\mathrm{Log}_{[V]}([V_*])) = \mathrm{Exp}_{[V_*]}((1-t)\,\mathrm{Log}_{[V_*]}([V]))$. This holds since both curves parametrize the unique length-minimizing geodesic from $[V]$ to $[V_*]$. $\quad\square$

*Proof of Corollary 2.* This is an immediate consequence of Proposition 1. In particular, it is enough to replace the occurrences of $g'(0)$ and $g''(0)$ with their expressions in terms of the gradient and Hessian of $F$ using (28) and (27) and using the coarse bounds

$$\frac{\sin^2(\theta)}{\theta^2} \geq \frac{4}{5}, \quad \psi(\theta) \leq \frac{3}{2},$$

valid for $\theta \in (0, \pi/4)$. $\quad\square$

## C  Technical lemmas and computations

This section collects several supporting lemmas and explicit computations used in the proofs of the main results in Section D. Throughout we make the assumption that $\mathrm{E}[\|X\|_2^2] < \infty$, so that $\Sigma$ is well-defined. Recall also the definition of $U_*^\perp$ from the second paragraph of Section 3.

### C.1  Gradient and Hessian computations

Define the function $\tilde{\ell} : \mathbb{R}^{d \times k} \times \mathbb{R}^d \to [0, \infty)$ by

$$\tilde{\ell}(U, x) := \frac{1}{2}\|x - UU^T x\|_2^2.$$

The reconstruction loss $\ell : \mathrm{Gr}(d, k) \times \mathbb{R}^d \to [0, \infty)$ is given by

$$\ell([U], x) := \tilde{\ell}(U, x), \tag{41}$$

which is well defined, as the right-hand side does not depend on the choice of representative in $[U]$. We thus have by (24) and (25), for any $[U] \in \mathrm{Gr}(d, k)$ and $\xi \in T_{[U]}\,\mathrm{Gr}(d, k)$

$$\mathrm{lift}_U(\mathrm{grad}\,\ell([U], x)) = -(I_d - UU^T)xx^T U, \tag{42}$$

$$\mathrm{lift}_U(\mathrm{Hess}\,\ell([U], x)[\xi]) = \Delta U^T xx^T U - (I_d - UU^T)xx^T \Delta, \tag{43}$$

where $\Delta = \mathrm{lift}_U(\xi)$, and where we computed the Euclidean gradient and Hessian of $\tilde{\ell}$ with respect to $U$ to obtain these expressions. We can express the empirical and population risk defined in Section 2 in terms of the reconstruction loss (41) as

$$R_n([U]) = \frac{1}{n}\sum_{i=1}^{n}\ell([U], X_i), \quad R([U]) = \mathrm{E}[\ell([U], X)].$$

By linearity of the grad and Hess operators along with (42) and (43), or alternatively by working with $\widetilde{R}$ and $\widetilde{R}_n$ defined in (2) and (1) and formulas (24) and (25), the gradients of $R$ and $R_n$ satisfy

$$\mathrm{grad}\,R([U]) = \mathrm{E}[\mathrm{grad}\,\ell([U], X))], \tag{44}$$

$$\mathrm{grad}\,R_n([U]) = \frac{1}{n}\sum_{i=1}^{n}\mathrm{grad}\,\ell([U], X_i), \tag{45}$$

and similarly for their Hessians

$$\mathrm{Hess}\,R([U])[\xi] = \mathrm{E}[\mathrm{Hess}\,\ell([U], X)[\xi]], \tag{46}$$

$$\mathrm{Hess}\,R_n([U])[\xi]) = \frac{1}{n}\sum_{i=1}^{n}\mathrm{Hess}\,\ell([U], X_i)[\xi]. \tag{47}$$

**Lemma 1.** *Let $i \in [d - k]$, $j \in [k]$, and $E_{ij} \in \mathbb{R}^{(d-k) \times k}$ be the matrix whose $(i, j)$-th entry is one and its remaining entries are zero. Define $\xi_{ij}$ by $\mathrm{lift}_{U_*}(\xi_{ij}) = U_*^\perp E_{i,j}$. Then*

$$\mathrm{Hess}\,R([U_*])[\xi_{ij}] = (\lambda_j - \lambda_{k+i}) \cdot \xi_{ij},$$

*i.e. $(\xi_{ij})$ are a basis of eigenvectors of $\mathrm{Hess}\,R([U_*])$ and their associated eigenvalues are $(\lambda_j - \lambda_{k+i})$.*

*Proof.* We have by (46) and (43)

$$\mathrm{lift}_{U_*}(\mathrm{Hess}\,R([U_*])[\xi_{ij}]) = U_*^\perp(E_{i,j}U_*^T \Sigma U_* - U_*^{\perp,T}\Sigma U_*^\perp E_{i,j})$$
$$= U_*^\perp(E_{i,j}\Lambda_{\leq k} - \Lambda_{>k}E_{i,j})$$
$$= (\lambda_j - \lambda_{k+i}) \cdot U_*^\perp E_{i,j}$$

where in the first line we used the identity $(I - U_*U_*^T) = U_*^\perp U_*^{\perp,T}$, and in the second we expanded $\Sigma = [U_* \mid U_*^T] \cdot \Lambda \cdot [U_* \mid U_*^T]^T$ and performed block-wise matrix multiplication. $\square$

**Corollary 3.** *Assume that $\lambda_k > \lambda_{k+1}$. Then* $\mathrm{Hess}\, R([U_*])$ *is positive definite, and for any $\xi \in T_{[U_*]}\mathrm{Gr}(d,k)$*

$$\mathrm{lift}_{U_*}(\mathrm{Hess}\, R([U_*])^{-1}[\xi]) = U_*^\perp C', \qquad \mathrm{lift}_{U_*}(\mathrm{Hess}\, R([U_*])^{-1/2}[\xi]) = U_*^\perp C'',$$

*where* $\mathrm{lift}_{U_*}(\xi) = U_*^\perp C$ *and*

$$C'_{ij} = \frac{C_{ij}}{\lambda_j - \lambda_{k+i}}, \qquad C''_{ij} = \frac{C_{ij}}{\sqrt{\lambda_j - \lambda_{k+i}}}.$$

*Proof.* The positive definiteness of $\mathrm{Hess}\, R([U_*])$ follows directly from the positiveness of its eigenvalues from Lemma 1 under the assumed condition $\lambda_k > \lambda_{k+1}$. This shows that $\mathrm{Hess}\, R([U_*])$ is invertible. Expanding $\xi$ into the basis of eigenvectors $(\xi_{ij})$ from Lemma 1 yields

$$\xi = \sum_{i=1}^{d-k}\sum_{j=1}^{k} \langle \xi, \xi_{ij}\rangle_{[U_*]} \cdot \xi_{ij} = \sum_{i=1}^{d-k}\sum_{j=1}^{k} \langle U_*^\perp C, U_*^\perp E_{ij}\rangle_F \cdot \xi_{ij} = \sum_{i=1}^{d-k}\sum_{j=1}^{k} C_{ij} \cdot \xi_{ij}.$$

where the second equality holds by (5). Therefore we have

$$\mathrm{Hess}\, R([U_*])^{-1}[\xi] = \sum_{i=1}^{d-k}\sum_{j=1}^{k} C_{ij}\, \mathrm{Hess}\, R([U_*])^{-1}[\xi_{ij}] = \sum_{i=1}^{d-k}\sum_{j=1}^{k} \frac{C_{ij}}{\lambda_j - \lambda_{k+i}} \cdot \xi_{ij}.$$

Applying $\mathrm{lift}_{U_*}$ to both sides and using its linearity yields the first identity of the corollary. The second follows from a similar argument. $\qquad\square$

### C.2 Convergence of empirical gradients and Hessians

**Lemma 2.** *For all $n \in \mathbb{N}$, it holds that*

$$\mathrm{E}[\|\mathrm{grad}\, R_n([U_*])\|_{H^{-1}}^2] = n^{-1} \cdot \mathrm{E}[\|\mathrm{grad}\, \ell([U_*], X)\|_{H^{-1}}^2],$$

*where $H = \mathrm{Hess}\, R([U_*])$ and where $\|\xi\|_{H^{-1}}^2 = \langle H^{-1}(\xi), \xi\rangle_{[U_*]}$ for $\xi \in T_{[U_*]}\mathrm{Gr}(d,k)$.*

*Proof.* By (45) we have

$$\mathrm{E}[\|\mathrm{grad}\, R_n([U_*])\|_{H^{-1}}^2] = \mathrm{E}\left[\left\|n^{-1}\sum_{i=1}^{n}\mathrm{grad}\, \ell([U_*], X_i)\right\|_{H^{-1}}^2\right]$$

$$= \frac{1}{n^2}\sum_{i=1}^{n}\sum_{j=1}^{n} \mathrm{E}[\langle H^{-1}[\mathrm{grad}\, \ell([U_*], X_i)], \mathrm{grad}\, \ell([U_*], X_j)\rangle_{[U_*]}]$$

$$= \frac{1}{n^2}\sum_{i=1}^{n}\mathrm{E}[\|\mathrm{grad}\, \ell([U_*], X_i)\|_{H^{-1}}^2],$$

and the statement follows since the elements of the sum are all equal to $\mathrm{E}[\|\mathrm{grad}\, \ell([U_*], X)\|_{H^{-1}}^2]$, since $(X_i)$ are i.i.d. with the same distribution as $X$. The third equality holds since the cross-terms vanish by the independence of $(X_i)$, $\mathrm{E}[\mathrm{grad}\, \ell([U_*], X)] = \mathrm{grad}\, R([U_*])$ by (44), and $\mathrm{grad}\, R([U_*]) = 0$ since $[U_*]$ is a minimizer of $R$. $\qquad\square$

**Lemma 3.** *It holds that*

$$\mathrm{E}[\|\mathrm{grad}\, \ell([U_*], X)\|_{H^{-1}}^2] = \sum_{i=1}^{d-k}\sum_{j=1}^{k} \frac{\mathrm{E}[\langle u_{k+i}, X\rangle^2 \langle u_j, X\rangle^2]}{\lambda_j - \lambda_{k+i}},$$

*where $H = \mathrm{Hess}\, R([U_*])$ and where $\|\xi\|_{H^{-1}}^2 = \langle H^{-1}(\xi), \xi\rangle_{[U_*]}$ for $\xi \in T_{[U_*]}\mathrm{Gr}(d,k)$.*

*Proof.* By (42), and using the identity $(I_d - U_* U_*^T) = U_*^\perp U_*^{\perp,T}$ we have

$$\mathrm{lift}_{U_*}(\mathrm{grad}\, \ell([U_*], X)) = -U_*^\perp[(U_*^{\perp,T}X)(U_* X)^T].$$

Let $C = (U_*^{\perp,T} X)(U_* X)^T$. It has entries $C_{i,j} = \langle u_{k+i}, X \rangle \cdot \langle u_j, X \rangle$. Hence by Corollary 3

$$\text{lift}_{U_*}(H^{-1}[\text{grad}\,\ell([U_*], X)]) = -U_*^{\perp} C',$$

where $C' = C_{ij}/(\lambda_j - \lambda_{k+i})$. Now

$$\text{E}[\|\text{grad}\,\ell([U_*], X)\|_{H^{-1}}^2] = \text{E}[\langle \text{lift}_{U_*}(H^{-1}[\text{grad}\,\ell([U_*], X)]), \text{lift}_{U_*}(\text{grad}\,\ell([U_*], X)) \rangle_F]$$

$$= \text{E}[\langle U_*^{\perp} C', U_*^{\perp} C \rangle_F] = \text{E}[\langle C', C \rangle_F] = \sum_{i=1}^{d-k} \sum_{j=1}^{k} \frac{\text{E}[C_{ij}^2]}{\lambda_j - \lambda_{k+i}}$$

Replacing $C_{ij}$ by its value yields the result. $\qquad\square$

**Lemma 4.** *Assume that for all $i, s \in [d-k]$ and $j, t \in [k]$*

$$\Lambda_{ijst} = \text{E}[\langle u_{k+i}, X \rangle \langle u_j, X \rangle \langle u_{k+s}, X \rangle \langle u_t, X \rangle] < \infty.$$

*Then*

$$\text{lift}_{U_*}(H_n^{-1}[\sqrt{n} \cdot \text{grad}\,R_n([U_*])] \xrightarrow{d} U_*^{\perp} G,$$

*where $H_n = \text{Hess}\,R_n([U_*])$, and where $G$ is a $(d-k) \times k$ matrix with jointly Gaussian mean zero entries with covariances $\text{E}[G_{ij} G_{st}] = \Lambda_{ijst}/(\delta_{ij}\delta_{st})$ where $\delta_{ij} = \lambda_j - \lambda_{k+i}$.*

*Proof.* Recall the global assumption $\text{E}\|X\|_2^2 < \infty$ so that the population covariance matrix $\Sigma$ exists. By (46) and (43) this implies that $H = \text{Hess}\,R([U_*])$ exists. Thus by (47) and the weak law of large numbers, we have $H_n \xrightarrow{P} H$, and by the continuous mapping theorem $H_n^{-1} \xrightarrow{P} H^{-1}$. On the other hand consider the random matrix

$$Z := U_*^{\perp,T} \text{lift}_{U_*}(H^{-1}[\text{grad}\,\ell([U_*], X)]).$$

By the proof of Lemma (3) we have $Z = -C'$ where $C'_{ij} = (\langle u_{k+i}, X \rangle \cdot \langle u_j, X \rangle)/(\lambda_j - \lambda_{k+i})$, thus $\text{E}[Z_{ij}] = 0$ and $\text{E}[Z_{ij} Z_{st}] = \Lambda_{ijst}/(\delta_{ij}\delta_{st})$. Hence by the central limit theorem and (45), as $n \to \infty$,

$$U_*^{\perp,T} \text{lift}_{U_*}(H^{-1}[\sqrt{n} \cdot \text{grad}\,R_n([U_*])]) = \frac{1}{\sqrt{n}} \cdot \sum_{i=1}^{n} Z_i \xrightarrow{d} G,$$

where $G$ is the Gaussian random matrix in the statement. Finally, by another application of the central limit theorem, we have that $\sqrt{n} \cdot \text{grad}\,R_n([U_*])$ converges in distribution to a random Gaussian element, and hence $(H_n^{-1} - H^{-1})[\sqrt{n} \cdot \text{grad}\,R_n([U_*])]$ converges to $0$ in probability by Slutsky's theorem. Therefore

$$U_*^{\perp,T} \text{lift}_{U_*}(H_n^{-1}[\sqrt{n} \cdot \text{grad}\,R_n([U_*])]$$
$$= \underbrace{U_*^{\perp,T} \text{lift}_{U_*}((H_n^{-1} - H^{-1})[\sqrt{n} \cdot \text{grad}\,R_n([U_*])])}_{\xrightarrow{P} 0} + \underbrace{U_*^{\perp,T} \text{lift}_{U_*}(H^{-1}[\sqrt{n} \cdot \text{grad}\,R_n([U_*])])}_{\xrightarrow{d} G},$$

and the final statement of the lemma is obtained by another application of Slutsky's theorem, an an application of the continuous mapping theorem with the map $C \mapsto U_*^{\perp} C$, recalling that $U_*^{\perp} U_*^{\perp,T} \Delta = \Delta$ for all $\Delta \in H_{U_*}$. $\qquad\square$

**Lemma 5.** *Assume that $\lambda_k > \lambda_{k+1}$ and that $\text{E}[X_j^4] < \infty$ for all $j \in [d]$. If*

$$n \geq 4(8\mathcal{V} + 1)\log(3k(d-k)) + 8(2\nu + 1)\log(4/\delta),$$

*then with probability at least $1 - \delta/4$,*

$$\lambda_{\min}(\widetilde{H}_n) \geq \frac{1}{2},$$

*where $\widetilde{H}_n = H^{-1/2} \circ H_n \circ H^{-1/2}$ and*

$$\mathcal{V} = \sup_{\|\xi\|=1} \text{E}[\|M[\xi]\|_{[U_*]}^2] - 1, \qquad \nu = \sup_{\|\xi\|=1} \text{E}[\|M^{1/2}[\xi]\|_{[U_*]}^4] - 1,$$

*where $M = H^{-1/2} \circ \text{Hess}\,\ell([U_*], X) \circ H^{-1/2}$, $H_n = \text{Hess}\,R_n([U_*])$, and $H = \text{Hess}\,R([U_*])$.*

*Proof.* Let $M_i = H^{-1/2} \circ \operatorname{Hess} \ell([U_*], X_i) \circ H^{-1/2}$. We have the variational characterization

$$1 - \lambda_{\min}(\widetilde{H}_n) = \lambda_{\max}(\operatorname{Id} - \widetilde{H}_n) = \sup_{\|\xi\|=1} \frac{1}{n} \sum_{i=1}^{n} (\operatorname{E}[\langle M[\xi], \xi \rangle_{[U_*]}] - \langle M_i[\xi], \xi \rangle_{[U_*]})$$

Thus by Bousquet's inequality [Bou02], specifically the version in [Van+16, Corollary 16.1], with probability at least $1 - \delta/4$

$$\lambda_{\max}(\operatorname{Id} - \widetilde{H}_n) \leq 2 \operatorname{E}[\lambda_{\max}(\operatorname{Id} - \widetilde{H}_n)] + \sqrt{\frac{2\nu \log(4/\delta)}{n}} + \frac{4\log(4/\delta)}{3n}.$$

Now by the Matrix Bernstein inequality [Tro+15, Theorem 6.6.1], we have

$$\operatorname{E}[\lambda_{\max}(\operatorname{Id} - \widetilde{H}_n)] = \operatorname{E}[\lambda_{\max}(\frac{1}{n} \sum_{i=1}^{n} \operatorname{Id} - M_i)] \leq \sqrt{\frac{2\mathcal{V} \log(3k(d-k))}{n}} + \frac{\log(3k(d-k))}{3n}.$$

Combining the two bounds and solving for $n$ yields the result. $\qquad\square$

**Lemma 6.** *The parameters $\mathcal{V}$ and $\nu$ defined in Lemma 5 admit the explicit expression given in Remark 3.*

*Proof.* Let $\xi \in T_{[U_*]} \operatorname{Gr}(d, k)$, and let $U_*^T C = \Delta = \operatorname{lift}_{U_*}(\xi)$. We have by Corollary 3, for $\xi_1 = H^{-1/2}[\xi]$

$$\operatorname{lift}_{U_*}(\xi_1) = U_*^\perp C',$$

where $C'_{ij} = C_{ij}/\sqrt{\lambda_j - \lambda_{k+i}}$. Now by (43), we have for $\xi_2 = \operatorname{Hess} \ell([U_*], X)[\xi_1]$

$$\operatorname{lift}_{U_*}(\xi_2) = U_*^\perp [C' U_*^T X X^T U_* - U_*^{\perp, T} X X^T U_*^\perp C']$$

Defining $\widetilde{X}_j = \langle X, u_j \rangle$, and writing $\widetilde{X}_{\leq k}$ for its first $k$ entries, and $\widetilde{X}_{>k}$ for its remaining $d-k$ entries, we have

$$\operatorname{lift}_{U_*}(\xi_2) = U_*^\perp [C' \widetilde{X}_{\leq k} \widetilde{X}_{\leq k}^T - \widetilde{X}_{>k} \widetilde{X}_{>k}^T C'].$$

Denote the term in brackets by $D$. Then again by Corollary 3, for $\xi_3 = H^{-1/2}[\xi_2]$, we get

$$\operatorname{lift}_{U_*}(\xi_3) = U_*^\perp D',$$

where $D'_{ij} = D_{ij}/\sqrt{\lambda_j - \lambda_{k+i}}$. We start with the computation of $\langle M[\xi], \xi \rangle_{[U_*]}$. We have

$$\begin{aligned}
\langle M[\xi], \xi \rangle_{[U_*]} &= \langle H^{-1/2} \circ \operatorname{Hess} \ell([U_*], X) \circ H^{-1/2}[\xi], \xi \rangle_{[U_*]} \\
&= \langle \operatorname{Hess} \ell([U_*], X) \circ H^{-1/2}[\xi], H^{-1/2}[\xi] \rangle_{[U_*]} \\
&= \langle C' \widetilde{X}_{\leq k} \widetilde{X}_{\leq k}^T - \widetilde{X}_{>k} \widetilde{X}_{>k}^T C', C' \rangle_F \\
&= \operatorname{Tr}(\widetilde{X}_{\leq k} \widetilde{X}_{\leq k}^T C'^T C') - \operatorname{Tr}(C'^T \widetilde{X}_{>k} \widetilde{X}_{>k}^T C') \\
&= \|C' \widetilde{X}_{\leq k}\|_2^2 - \|C'^T \widetilde{X}_{>k}\|_2^2 \\
&= \sum_{i=1}^{d-k} \left( \sum_{j=1}^{k} \frac{C_{ij}}{\lambda_j - \lambda_{k+i}} \cdot \widetilde{X}_j \right)^2 - \sum_{j=1}^{k} \left( \sum_{i=1}^{d-k} \frac{C_{ij}}{\lambda_j - \lambda_{k+i}} \widetilde{X}_{k+i} \right)^2
\end{aligned}$$

Taking the square of this expression, expanding, and taking expectations yields an explicit expression of $\langle M[\xi], \xi \rangle_{[U_*]}^2$ in terms of $C$. Noting that the map that sends $\xi$ to $C$ is an isometric isomorphism, the supremum of the former over vectors $\|\xi\| = 1$ is equal to the supremum of the latter over $\|C\|_F = 1$. This concludes the proof for $\nu$. For $\mathcal{V}$, note that

$$\begin{aligned}
\|M[\xi]\|_{[U_*]}^2 &= \|U_*^\perp D\|_F^2 = \|D\|_F^2 \\
&= \sum_{i=1}^{d-k} \sum_{j=1}^{d-k} \frac{1}{\lambda_j - \lambda_{k+i}} \cdot \left( \widetilde{X}_j \sum_{s=1}^{k} \frac{C_{is}}{\sqrt{\lambda_s - \lambda_{k+i}}} \widetilde{X}_s - \widetilde{X}_{k+i} \sum_{t=1}^{d-k} \frac{C_{tj}}{\sqrt{\lambda_j - \lambda_{k+t}}} \widetilde{X}_{k+t} \right)^2
\end{aligned}$$

Expanding the square and taking expectations gives an explicit expression of $\|M[\xi]\|_2^2$ in terms of $C$, and by the same argument as for $\nu$, $\mathcal{V}$ is the supremum over $\|C\|_F = 1$ of this explicit expression. $\qquad\square$

## C.3 Localization argument for ERM

The following two corollaries are immediate consequences of Corollary 2 and Proposition 2, since the population and empirical reconstruction risks $R$ and $R_n$ are, up to an insignificant constant, equal to the negative block Rayleigh quotients $-(1/2)\operatorname{Tr}(U^T\Sigma U)$ and $-(1/2)\operatorname{Tr}(U^T\Sigma_n U)$ respectively.

**Corollary 4.** *Assume that $\lambda_k > \lambda_{k+1}$, and let $[U] \in \operatorname{Gr}(d,k)$ such that $\theta_k([U_*],[U]) < \pi/4$. Then*

$$R_n([U]) - R_n([U_*]) \geq \langle \operatorname{grad} R_n([U_*]), \xi \rangle_{[U_*]} + \frac{2}{5}\langle \operatorname{Hess} R_n([U_*])[\xi], \xi \rangle_{[U_*]}$$

$$R([U]) - R([U_*]) \leq \frac{3}{4}\langle \operatorname{Hess} R([U_*])[\xi], \xi \rangle_{[U_*]}$$

*where $\xi = \operatorname{Log}_{[U_*]}([U])$.*

**Corollary 5.** *It holds that*

$$\sup_{[U]\in\operatorname{Gr}(d,k)} \sup_{\|\xi_1\|=1, \|\xi_2\|=1, \|\xi_3\|=1} |\nabla^3 R([U], \xi_1, \xi_2, \xi_3)| \leq 4\|\Sigma\|_F,$$

*and for all $[U] \in \operatorname{Gr}(d,k)$ and $\xi \in T_{[U]}\operatorname{Gr}(d,k)$,*

$$\|P_{\xi,1}^{-1} \circ \operatorname{Hess} R_n(\operatorname{Exp}_{[U]}(\xi)) \circ P_{\xi,1} - \operatorname{Hess} R_n([U])\|_{\operatorname{op}} \leq 4\|\Sigma_n\|_F\|\xi\|.$$

**Lemma 7.** *On the event that $\theta_k([U_n],[U_*]) < \pi/4$, it holds that*

$$\|\operatorname{Log}_{[U_*]}([U_n])\|_H^2 \leq \frac{25}{4} \cdot \lambda_{\min}^{-2}(\widetilde{H}_n) \cdot \|\operatorname{grad} R_n([U_*])\|_{H^{-1}}^2,$$

*where $\widetilde{H}_n = H^{-1/2} \circ H_n \circ H^{-1/2}$, $\lambda_{\min}(\widetilde{H}_n)$ is its smallest eigenvalue, $H = \operatorname{Hess} R([U_*])$, $H_n = \operatorname{Hess} R_n([U_*])$, and $\|\xi\|_H^2 = \langle H(\xi), \xi \rangle_{[U_*]}$ for $\xi \in T_{[U_*]}\operatorname{Gr}(d,k)$.*

*Proof.* Denote $\operatorname{Log}_{[U_*]}([U_n])$ by $\xi_n$. Since $[U_n]$ minimizes the empirical risk we have

$$R_n([U_n]) - R_n([U_*]) \leq 0. \tag{48}$$

On the other hand by Corollary 4 we have

$$R_n([U_n]) - R_n([U_*]) \geq \langle \operatorname{grad} R_n([U_*]), \xi_n \rangle + \frac{2}{5}\|\xi_n\|_{H_n}^2. \tag{49}$$

By the Cauchy-Schwartz inequality

$$\langle \operatorname{grad} R_n([U_*]), \xi_n \rangle \geq -\|\operatorname{grad} R_n([U_*])\|_{H^{-1}} \cdot \|\xi_n\|_H. \tag{50}$$

And we also have

$$\|\xi_n\|_{H_n}^2 \geq \left(\inf_{\xi\neq 0} \frac{\|\xi\|_{H_n}^2}{\|\xi\|_H^2}\right) \cdot \|\xi_n\|_H^2 = \inf_{\|\xi\|=1}\langle \widetilde{H}_n[\xi], \xi \rangle \cdot \|\xi_n\|_H^2 = \lambda_{\min}(\widetilde{H}_n) \cdot \|\xi_n\|_H^2. \tag{51}$$

Combining (48), (49), (50), and (51) we obtain the result. $\quad\square$

## C.4 Matrix identities and perturbation bounds

**Lemma 8.** *Assume that $\lambda_k - \lambda_{k+1} > 0$ and that $\|\Sigma - \Sigma_n\|_{\operatorname{op}} \leq (\lambda_k - \lambda_{k+1})/2$. Then*

$$\sin\theta_k([U_n],[U_*]) \leq \frac{2\|\Sigma_n - \Sigma\|_{\operatorname{op}}}{\lambda_k - \lambda_{k+1}}.$$

*Proof.* Let $\delta = \lambda_k - \lambda_{k+1}$. Recall that $(u_{n,j})$ is an orthonormal basis of eigenvectors of $\Sigma_n$ ordered non-increasingly according to their corresponding eigenvalues $(\lambda_{n,j})$, with ties broken arbitrarily. On the one hand, we have by inequality (1.63) in [Tao12]

$$|\lambda_j - \lambda_{j,n}| \leq \|\Sigma_n - \Sigma\|_{\operatorname{op}}, \tag{52}$$

for all $j \in [d]$. This implies that

$$\lambda_{n,k+1} \leq \lambda_{k+1} + \|\Sigma_n - \Sigma\|_{\operatorname{op}} < \lambda_{k+1} + (\lambda_k - \lambda_{k+1}) \leq \lambda_k, \tag{53}$$

where the inequality follows by the assumed bound in the lemma. On the other hand, we have by the operator norm version of Theorem 1 in [YWS15][3], and taking $r = 1$ and $s = k$ in the statement

$$\sin\theta_k([U_n], [U_*]) \leq \frac{\|\Sigma_n - \Sigma\|_{\mathrm{op}}}{\lambda_k - \lambda_{n,k+1}},$$

where we have used (53) to simplify the denominator appearing in the original statement. Using again (52) to lower bound the denominator we get

$$\sin\theta_k([U_n], [U_*]) \leq \frac{\|\Sigma_n - \Sigma\|_{\mathrm{op}}}{\delta - \|\Sigma_n - \Sigma\|_{\mathrm{op}}}.$$

Using the inequality $x/(c - x) \leq 2x/c$ valid for $x \in [0, c/2]$ finishes the proof. $\qquad\square$

**Corollary 6.** *Assume that $\lambda_k > \lambda_{k+1}$ and that $\mathrm{E}[X_j^4] < \infty$ for all $j \in [d]$. If*

$$n \geq \frac{16(\mathcal{S} + r(n))}{\delta(\lambda_k - \lambda_{k+1})^2},$$

*Then with probability at least $1 - \delta/2$*

$$\theta_k([U_n], [U_*]) < \pi/4,$$

*where*

$$\mathcal{S} := c(d) \cdot \|\mathrm{E}[(XX^T - \Sigma)^2]\|_{\mathrm{op}}, \qquad r(n) := c^2(d) \cdot n^{-1} \mathrm{E}[\max_{i \in [n]} \|X_i X_i^T - \Sigma\|_{\mathrm{op}}^2],$$

*and $c(d) = 4(1 + 2\lceil \log(d) \rceil)$.*

*Proof.* By the second item of Theorem 5.1 in [Tro16], we have

$$n \cdot \mathrm{E}[\|\Sigma_n - \Sigma\|_{\mathrm{op}}^2] \leq 2\mathcal{S} + 2r(n)$$

Applying Markov's inequality, using Lemma 8, and solving for $n$ yields the result. $\qquad\square$

**Lemma 9.** *Let $U, V \in \mathrm{St}(d, k)$. Let $(\sigma_i)_{i=1}^d$ be the singular values of $UU^T - VV^T$ ordered non-increasingly. Then for $i \in [2k]$,*

$$\sigma_i = \sin\theta_{k - \lfloor (i-1)/2 \rfloor}([U], [V]),$$

*and the remaining singular values are zero.*

*Proof.* See Theorem 5.5 in Chapter 1 of [SS90]. $\qquad\square$

# D  Proofs of main results

## D.1  Proof of Theorem 1

**Consistency.** Since $\mathrm{E}[\|X\|_2^2] < \infty$, we have by the weak law of large numbers $\Sigma_n \xrightarrow{p} \Sigma$, i.e. for all $\varepsilon > 0$

$$\lim_{n \to \infty} \mathrm{P}(\|\Sigma_n - \Sigma\|_{\mathrm{op}} \geq \varepsilon) = 0.$$

Let $A_n$ be the event that $\|\Sigma_n - \Sigma\|_{\mathrm{op}} < (\lambda_k - \lambda_{k+1})/2$. By assumption the right-hand side is strictly larger than 0, so $\lim_{n \to \infty} \mathrm{P}(A_n) = 1$. On the one hand, by Lemma 8, we have on the event $A_n$

$$\sin(\theta_k([U_n], [U_*])) \leq \frac{2\|\Sigma_n - \Sigma\|_{\mathrm{op}}}{\lambda_k - \lambda_{k+1}}.$$

On the other we have

$$\mathrm{dist}([U_n], [U_*]) = \left( \sum_{j=1}^k \theta_j^2([U_n], [U_*]) \right)^{1/2} \leq \sqrt{k} \cdot \theta_k([U_n], [U_*]).$$

---

[3]See the beginning of the second paragraph after the statement of the theorem.

Now let $0 < \varepsilon < \sqrt{k}\pi/2$. Using the above two bounds we obtain

$$
\begin{aligned}
\mathrm{P}(\mathrm{dist}([U_n],[U_*]) \geq \varepsilon) &\leq \mathrm{P}(\theta_k([U_n],[U_*]) \geq \varepsilon/\sqrt{k}) \\
&\leq \mathrm{P}(\{\theta_k([U_n],[U_*]) \geq \varepsilon/\sqrt{k}\} \cap A_n) + \mathrm{P}(A_n^c) \\
&\leq \mathrm{P}(\|\Sigma_n - \Sigma\|_{\mathrm{op}} \geq (\lambda_k - \lambda_{k+1}) \cdot \sin(\varepsilon/\sqrt{k})/2) + \mathrm{P}(A_n^c)
\end{aligned}
$$

and both terms go to zero as $n \to \infty$.

$\sqrt{n}$**-consistency.** For the rest of the proof, let $\xi_n = \mathrm{Log}_{[U_*]}([U_*])$ when well defined, and $0$ otherwise. Let $H = \mathrm{Hess}\,R([U_*])$ and $H_n = \mathrm{Hess}\,R_n([U_*])$, and for a positive semi-definite linear map $L : T_{[U_*]}\,\mathrm{Gr}(d,k) \to T_{[U_*]}\,\mathrm{Gr}(d,k)$, let $\|\xi\|_L^2 = \langle L(\xi), \xi \rangle$ denote the (squared) semi-norm it induces, for $\xi \in T_{[U_*]}\,\mathrm{Gr}(d,k)$. We note that under the eigengap assumption of the theorem, $\xi \mapsto \|\xi\|_H$ is a norm since $H$ is positive-definite by Corollary 3.

Let $A_n$ be the event that $\theta_k([U_n],[U_*]) < \pi/4$. By the consistency result, $\lim_{n\to\infty} \mathrm{P}(A_n) = 1$. Now on this event we have by Lemma 7

$$
\|\xi_n\|_H \leq \frac{5}{2} \cdot \lambda_{\min}^{-1}(\widetilde{H}_n) \cdot \|\mathrm{grad}\,R_n([U_*])\|_{H^{-1}},
$$

where $\widetilde{H}_n = H^{-1/2} \circ H_n \circ H^{-1/2}$ and $\lambda_{\min}(\widetilde{H}_n)$ is its smallest eigenvalue.

On the one hand, by the weak law of large numbers $\widetilde{H}_n \xrightarrow{p} \mathrm{Id}$, and by the continuous mapping theorem $\lambda_{\min}^{-1}(\widetilde{H}_n) \xrightarrow{p} 1$. Let $B_n$ be the event that $\lambda_{\min}^{-1}(\widetilde{H}_n) \leq 2$, which thus satisfies $\lim_{n\to\infty} \mathrm{P}(B_n) = 1$.

On the other hand, we have by Lemma 2

$$
\mathrm{E}[\|\mathrm{grad}\,R_n([U_*])\|_{H^{-1}}^2] = n^{-1}\,\mathrm{E}[\|\mathrm{grad}\,\ell([U_*], X)\|_{H^{-1}}^2],
$$

and by the moment assumption of the theorem and Lemma 3 the expectation on the right-hand side is finite.

Therefore we have for any $\varepsilon > 0$ and $\alpha \in [0, 1/2)$, using the above two displays and Markov's inequality

$$
\begin{aligned}
\mathrm{P}(\|\xi_n\|_H \geq \frac{\varepsilon}{n^\alpha}) &\leq \mathrm{P}(\{\|\xi_n\|_H \geq \frac{\varepsilon}{n^\alpha}\} \cap (A_n \cap B_n)) + \mathrm{P}(A_n^c) + \mathrm{P}(B_n^c) \\
&\leq \mathrm{P}(\|\mathrm{grad}\,R_n([U_*])\|_{H^{-1}} \geq \frac{\varepsilon}{5n^\alpha}) + \mathrm{P}(A_n^c) + \mathrm{P}(B_n^c) \\
&\leq 25 \cdot n^{2\alpha - 1} \cdot \varepsilon^{-2} \cdot \mathrm{E}[\|\mathrm{grad}\,\ell([U_*], X)\|_{H^{-1}}^2] + \mathrm{P}(A_n^c) + \mathrm{P}(B_n^c)
\end{aligned}
$$

and all three terms go to zero as $n \to \infty$ so that for all $\varepsilon > 0$ and $\alpha \in [0, 1/2)$

$$
\lim_{n\to\infty} \mathrm{P}(\|\xi_n\|_H \geq \frac{\varepsilon}{n^\alpha}) = 0. \tag{54}
$$

**Asymptotic normality.** Let $A_n$ be the event that $\theta_k([U_n],[U_*]) < \pi/4$. By the consistency result, $\lim_{n\to\infty} \mathrm{P}(A_n) = 1$.

Since $[U_n]$ minimizes the empirical reconstruction risk, it holds that $\mathrm{grad}\,R_n([U_n]) = 0$. On the event $A_n$, we have by the Taylor expansion (30)

$$
0 = \mathrm{grad}\,R_n([U_n]) = \mathrm{grad}\,R_n([U_*]) + H_n[\xi_n] + E_n,
$$

where we used that the parallel transport map introduced at the beginning of Appendix A sends 0 to 0, and where the term $E_n$ is given by

$$
E_n = \int_0^1 \{P_{\xi_n,s}^{-1} \circ \mathrm{Hess}\,R_n(\mathrm{Exp}_{[U_*]}(s\xi_n)) \circ P_{\xi_n,s} - \mathrm{Hess}\,R_n([U_*])\}[\xi_n]\,\mathrm{d}s.
$$

By the weak law of large numbers $H_n \xrightarrow{p} H$ which is invertible by the eigengap assumption of the theorem and Corollary 3. The event $B_n$ on which $H_n$ is invertible thus satisfies $\lim_{n\to\infty} \mathrm{P}(B_n) = 1$. On the event $A_n \cap B_n$ we thus have, rearranging the first display and scaling by $\sqrt{n}$

$$
\sqrt{n} \cdot \xi_n = -H_n^{-1}[\sqrt{n} \cdot \mathrm{grad}\,R_n([U_*])] - H_n^{-1}[\sqrt{n} \cdot E_n]
$$

Now we claim that $\sqrt{n} \cdot E_n \xrightarrow{p} 0$. Indeed, we have by Corollary 5

$$\|E_n\| \le \|\xi_n\| \cdot \int_0^1 \|P_{s\xi_n,1}^{-1} \circ \mathrm{Hess}\, R_n(\mathrm{Exp}_{[U_*]}(s\xi_n)) \circ P_{s\xi_n,1} - \mathrm{Hess}\, R_n([U_*])\|_{\mathrm{op}} \mathrm{d}s$$

$$\le 2 \cdot \|\xi_n\|^2 \cdot \|\Sigma_n\|_F.$$

where in the first line we used the easy to check identity $P_{\xi,s} = P_{s\xi,1}$. Once again using the weak law of large numbers and the continuous mapping theorem we obtain $\|\Sigma_n\|_F \xrightarrow{p} \|\Sigma\|_F$. Thus using (54) we obtain $\sqrt{n} \cdot E_n \xrightarrow{p} 0$. The asymptotic normality statement in the Theorem then follows from Lemma 4, Slutsky's theorem, and the fact that the events $A_n^c$ and $B_n^c$ have vanishing probability as $n \to \infty$.

**Excess Risk.** Let $A_n$ be the event that $\theta_k([U_n], [U_*]) < \pi/4$. By the consistency result, $\lim_{n\to\infty} \mathrm{P}(A_n) = 1$. By the Taylor expansion (29), we have on this event

$$n \cdot [R([U_n]) - R([U_*])] = \frac{1}{2}\|\sqrt{n} \cdot \xi_n\|_H^2 + n \cdot E_n,$$

where the error term $E_n$ is given by

$$E_n = \frac{s^3}{6}\nabla^3 R(\mathrm{Exp}_{[U_*]}(\xi_n), P_{s\xi_n,1}(\xi_n), P_{s\xi_n,1}(\xi_n), P_{s\xi_n,1}(\xi_n)),$$

for some $s \in [0,1]$, and where we again used the easy to check identity $P_{\xi,s} = P_{s\xi,1}$. We claim that $n \cdot E_n \xrightarrow{p} 0$. Indeed by Corollary 5, we have

$$\|E_n\| \le \|\Sigma\|_F \cdot \|P_{s\xi_n,1}(\xi_n)\|^3 = \|\Sigma\|_F \cdot \|\xi_n\|^3,$$

where we used in the equality that the parallel transport map is an isometry as described in the beginning of Appendix A. Thus using (54) we obtain $n \cdot E_n \xrightarrow{p} 0$. Finally, we have, using the asymptotic normality statement in the theorem, proven above, the continuous mapping theorem, and the explicit description of $H^{1/2}$ from Lemma 1 that $\mathrm{lift}_{U_*}(H^{1/2}[\sqrt{n} \cdot \xi_n])$ converges in distribution $U_*^\perp H$ where $H$ is the Gaussian matrix in the statement of the theorem. The result then follows from an application of Slutsky's theorem, the fact that the event $A_n^c$ has vanishing probability as $n \to \infty$, and that $\|U_*^\perp C\|_F = \|C\|_F$ for any $(d-k) \times k$ matrix $C$.

## D.2 Proof of Remark 1

We have on the one hand by Lemma 9 that

$$\|U_n U_n^T - U_* U_*^T\|_p = 2^{1/p} \cdot \Big(\sum_{j=1}^k \sin^p(\theta_j)\Big)^{1/p}, \tag{55}$$

where we have shortened $\theta_j = \theta_j([U_n], [U_*])$. Recall from Section 2 that the singular values of $\Delta_n = \mathrm{lift}_{U_*}(\mathrm{Log}_{[U_*]}([U_n]))$ are the principal angles between $[U_n]$ and $[U_*]$. Define the function $\varphi : \mathbb{R}^{d \times k} \to \mathbb{R}^{d \times k}$ as follows. For a matrix $A \in \mathbb{R}^{d \times k}$, let $A = PSQ^T$ be a SVD of $A$. Then we define $\varphi(A) = P\sin(S)Q^T$ where $\sin$ is applied element-wise to the singular values. Now

$$\varphi(A) - \varphi(0) - A = P\sin(S)Q^T - PSQ^T = P(\sin(S) - S)Q^T,$$

Hence

$$\lim_{\|A\|_{\mathrm{op}} \to 0} \frac{\|\varphi(A) - \varphi(0) - A\|_{\mathrm{op}}}{\|A\|_{\mathrm{op}}} \le \lim_{\|A\|_{\mathrm{op}} \to 0} \frac{s_{\max} - \sin(s_{\max})}{s_{\max}} = 0$$

where $s_{\max}$ is the maximum singular value of $A$, and we used the fact that $(x - \sin(x))/x \to 0$ as $x \to 0$. Therefore $\varphi$ is differentiable at 0, and its derivative there is the identity map, so by the delta method [e.g. Van00, Theorem 3.1]) and the asymptotic normality result in Theorem 1 we obtain

$$\sqrt{n} \cdot \varphi(\Delta_n) \xrightarrow{d} U_*^\perp G$$

Applying the continuous mapping theorem with the map $A \mapsto 2^{1/p}\|A\|_p$, using (55), and noting that $\|U_*^\perp G\|_p = \|G\|_p$ since $U_*^\perp \in \mathrm{St}(d, d-k)$ finishes the proof.

## D.3  Proof of Theorem 2

Let $\xi_n = \mathrm{Log}_{[U_*]}([U_*])$ when well defined, and $0$ otherwise. Let $H = \mathrm{Hess}\, R([U_*])$ and $H_n = \mathrm{Hess}\, R_n([U_*])$, and for a positive semi-definite linear map $L : T_{[U_*]} \mathrm{Gr}(d,k) \to T_{[U_*]} \mathrm{Gr}(d,k)$, let $\|\xi\|_L^2 = \langle L(\xi), \xi \rangle$ denote the (squared) semi-norm it induces, for $\xi \in T_{[U_*]} \mathrm{Gr}(d,k)$.

Let $A_n$ be the event that $\theta_k([U_n], [U_*]) < \pi/4$. Under the sample size restriction of the theorem, and in particular the third term in this restriction, this event happens with probability at least $1 - \delta/2$ by Corollary 6.

Now on this event, we have by Lemma 7 that the following inequality holds

$$\|\xi_n\|_H \le \frac{5}{2} \cdot \lambda_{\min}^{-1}(\widetilde{H}_n) \cdot \|\mathrm{grad}\, R_n([U_*])\|_{H^{-1}},$$

where $\widetilde{H}_n = H^{-1/2} \circ H_n \circ H^{-1/2}$.

Let $B_n$ be the event that $\lambda_{\min}(\widetilde{H}_n) \ge 1/2$. Under the sample size restriction of the theorem, and in particular the first two terms in this restriction, this event happens with probability at least $1 - \delta/4$ by Lemmas 5 and 6.

Now by Lemmas 2 and 3 and Markov's inequality we also have that on an event $C_n$ that holds with probability at least $1 - \delta/4$

$$\|\mathrm{grad}\, R_n([U_*])\|_{H^{-1}}^2 \le \frac{4}{n \cdot \delta} \cdot \sum_{i=1}^{d-k} \sum_{j=1}^{k} \frac{\mathrm{E}[\langle u_{k+i}, X \rangle^2 \langle u_j, X \rangle^2]}{\lambda_j - \lambda_{k+i}}.$$

Therefore, on the event $A_n \cap B_n \cap C_n$ that holds with probability at least $1 - \delta$, we have

$$\|\xi_n\|_H^2 \le \frac{100}{n \cdot \delta} \sum_{i=1}^{d-k} \sum_{j=1}^{k} \frac{\mathrm{E}[\langle u_{k+i}, X \rangle^2 \langle u_j, X \rangle^2]}{\lambda_j - \lambda_{k+i}}$$

and on this same event, we have by Corollary 4, noting that on this event $\xi_n = \mathrm{Log}_{[U_*]}([U_*])$

$$R([U_n]) - R([U_*]) \le \frac{75}{n \cdot \delta} \sum_{i=1}^{d-k} \sum_{j=1}^{k} \frac{\mathrm{E}[\langle u_{k+i}, X \rangle^2 \langle u_j, X \rangle^2]}{\lambda_j - \lambda_{k+i}}.$$

which concludes the proof.

## D.4  Proof Sketch for Example 2

When $X \sim \mathcal{N}(0, \Sigma)$, we have

$$\Gamma_{jrsp} = \begin{cases} 3\lambda_j^2 & \text{if } j = s = r = p \\ \lambda_j \lambda_r & \text{if } j = p \ne r = s \text{ or } j = s \ne r = p \\ \lambda_j \lambda_s & \text{if } j = r \ne s = p \\ 0 & \text{otherwise} \end{cases}$$

and

$$\Lambda_{tsqr} = \begin{cases} \lambda_{k+t} \lambda_s & \text{if } t = q \text{ and } s = r \\ 0 & \text{otherwise} \end{cases}$$

and finally

$$\Omega_{itql} = \begin{cases} 3\lambda_{k+i}^2 & \text{if } i = t = q = l \\ \lambda_{k+i} \lambda_{k+t} & \text{if } i = q \ne t = l \text{ or } i = l \ne t = q \\ \lambda_{k+i} \lambda_{k+1q} & \text{if } i = t \ne q = l \\ 0 & \text{otherwise} \end{cases}$$

Using these identities in the sums in Remark 3 and simplifying shows that the maximization problem becomes one over the simplex which can be solved directly for $\mathcal{V}$ or very well-approximated for $\nu$.

### D.5 Remarks on omitted proofs

**Quantile bounds in Corollary 1 and Remark 1.** The upper bounds in these statements are a direct consequence of the Gaussian concentration inequality [e.g. BLM13, Theorem 5.6]. For the lower bounds and a step by step derivation, see for example [EME24, Appendix A]. Note also that the quantile bounds in Example 1 are a direct consequence of those in Corollary 1 and Remark 1.

**Claim in Remark 2.** As is clear from the Proof of Theorem 1, the key properties leading to it are the self-concordance and Hessian Lipschitzness of the empirical and populations reconstruction risks, i.e. Corollaries 4 and 5. These are themselves derived from the more general results we obtained in Propositions 1 and 2 that holds for the negative block Rayleigh quotient. These can be used directly with minor adjustments to prove the claim we made in Remark 2. Furthermore, up to generalizing the parameters appearing in the sample size restriction, one may use Propositions 1 and 2 to obtain a more general version of Theorem 2 that holds in the setting described in Remark 2.

