# OpenReview forum: "A Geometric Analysis of PCA"
_NeurIPS.cc/2025/Conference — NeurIPS 2025 poster_

### Official Review · Reviewer_PSP6 · 2025-06-28

**Clarity:** 2
**Significance:** 2
**Originality:** 2
**Rating:** 4
**Confidence:** 2

**Summary:**

This paper presents a rigorous and unified theoretical analysis of the excess risk of PCA given the property of data distribution. Key contributions include: (1) a central limit theorem for the error of the principal subspace estimated by PCA; (2) the negative block Rayleigh quotient, defined on the Grassmannian, is generalized self-concordant along geodesics emanating from its minimizer of maximum rotation less than $\pi/4$; (3) derive the asymptotic distribution of PCA’s excess risk under the reconstruction loss.

**Questions:**

1. Can the authors provide simulations to validate the non-asymptotic bound and compare it to existing empirical estimates?
2. While the analysis is not directly applicable to infinite-dimensional settings, could a kernelized variant be derived under similar geometric assumptions?
3. Have the authors considered any improvement of the mentioned limitations?

**Ethical Concerns:**

["NO or VERY MINOR ethics concerns only"]

**Limitations:**

Yes.

**Paper Formatting Concerns:**

No.

**Quality:**

3

**Strengths And Weaknesses:**

Strengths:

1.	The main results are clearly stated and proved.
2.	The two examples are good to help readers understand the application of results.
3.	The use of Grassmannian geometry, self-concordance theory, and a second-order Taylor expansion for loss functions on manifolds is powerful.

Weaknesses:

1.	The paper does not include experiments to validate the theoretical bounds or show their practical performance, even on synthetic or simulated data.
2.	Two limitations mentioned in the paper.
3.	The analysis does not yet extend to Kernel PCA, functional PCA, or high-dimensional regimes, which are increasingly important.
4.	The authors assume a high level of familiarity with Riemannian geometry and asymptotic statistics. While technically sound, it may be challenging for a general machine learning audience to follow.

---

> ### Author Rebuttal · Authors · 2025-07-31
>
> We thank the reviewer for reviewing our paper and for the questions.
>
> ***Can the authors provide simulations to validate the non-asymptotic bound and compare it to existing empirical estimates?***
>
> We will add a section containing results of simulations that compare our predictions with empirically measured quantities. Please note however that while such simulations help illustrate our results, they cannot validate them - the simulations can necessarily only be run on a handful of data distributions, whereas our claims are made for general data distributions.
>
>
> ***While the analysis is not directly applicable to infinite-dimensional settings, could a kernelized variant be derived under similar geometric assumptions?***
>
> As long as the dimension of the RKHS is finite dimensional, our results apply. As we have explained in the paper, addressing the infinite dimensional case remains out of reach using our approach: we would need to rigorously define an "infinite dimensional Grassmannian" and all its associated geometric objects.
>
> If we misunderstood the reviewer's question, could they please clarify it?
>
>
> ***Have the authors considered any improvement of the mentioned limitations?***
>
> We have tried overcoming them but did not succeed. The discussion section outlines where we believe the difficulty lies in overcoming these limitations.

---

### Official Review · Reviewer_fztr · 2025-06-30

**Clarity:** 3
**Significance:** 2
**Originality:** 2
**Rating:** 4
**Confidence:** 3

**Summary:**

The paper studies the excess risk of PCA. Using tools from Riemannian manifolds and self-concordant analysis, it derives an exact expression for the asymptotic excess risk and provides a non-asymptotic upper bound for the excess risk of PCA. The paper also establishes the asymptotic distribution of a principal subspace-related quantity.

**Questions:**

1. Is there an explicit expression for the term $lift_{U_*}(\log_{[U_*]}([U_n]))$ ? It is difficult to interpret the term without an explicit expression.
2. Working with projection matrices directly also avoids the ambiguity in $U$ and, in my view, leads to cleaner and more interpretable results. Is there a particular reason the authors chose not to use the projection formulation directly?

**Ethical Concerns:**

["NO or VERY MINOR ethics concerns only"]

**Final Justification:**

**Resolved issues:** the authors now provide an intuitive explanation of quantities such as $lift_{U_*}(\log_{[U_*]}([U_n]))$, which makes the paper clearer. They also commit to adding numerical results. Based on the discussion among the other reviewers, the technical assumptions appear mild and do not seem unsuitable from my perspective.
**Unresolved issues": Many of the results appear to be analyzable using the projective matrix approach, which does not seem to require Riemannian manifold tools and may be more accessible to a general audience. As I am not very familiar with Riemannian manifold theory, it is difficult for me to assess whether the Riemannian manifold perspective introduces genuinely new tools or yields fundamentally novel results. Other reviewers have also expressed similar doubts regarding the novelty of the results.

Based on the above two points, I maintain my recommended score.

**Limitations:**

yes

**Paper Formatting Concerns:**

No concern.

**Quality:**

3

**Strengths And Weaknesses:**

**Strengths:**

The authors provide a detailed theoretical analysis of the excess risk of PCA using tools from Riemannian geometry, offering an interesting framework for studying eigenspace and singular subspace perturbations as well as the excess risk of spectral methods. Notably, their asymptotic results require only mild moment conditions on $X$. I am not familiar with Riemannian manifolds , so I cannot check the correctness of the results, but the paper appears to be well-written.

**Weaknesses:**

1. The results in Theorem 1 are not surprising. A closely related asymptotic result for the excess risk appears in Proposition 2.14 of Non-Asymptotic Upper Bounds for the Reconstruction Error of PCA by Reiss and Wahl (2020). Although their result is stated under a Gaussian assumption, the proof appears to use Gaussian primarily to ensure full-rank conditions and distinct eigenvalues, suggesting that the result extends beyond the Gaussian setting. The authors should clarify how their results differ in a substantive way from those in Reiss and Wahl (2020).
2. The authors do not provide numerical results to support their theoretical findings. Including a few synthetic experiments would help illustrate how to interpret the theoretical results and would also serve to validate their theoretical claims.

---

> ### Author Rebuttal · Authors · 2025-07-31
>
> We thank the reviewer for reviewing our paper. We address their questions and comments below.
>
> ***Is there an explicit expression for the term $lift_{U_{\*}}(Log_{[U_{\*}]}([U_{n}]))$? It is difficult to interpret the term without an explicit expression.***
>
> There is indeed an explicit formula for $lift_{U_{\*}}(Log_{[U_{\*}]}([U_{n}]))$ in terms of the matrices $U_{\*}$ and $U_{n}$ - it is described in the paragraph "Logarithmic map" in Section 2.1. Would it help the reader to restate the explicit formula found in Section 2.1 in the vicinity of Theorem 1?
>
> Intuitively however, this term is most easily viewed as a rigorous version of "$[U_{n}] - [U_{\*}]$"; as the Riemannian analogue of $\mu_n - \mu_{\*}$: the vector in the classical multivariate CLT setting that takes one from the mean $\mu_{\*}$ to the sample mean $\mu_n$.
>
> ***Working with projection matrices directly also avoids the ambiguity in $U$ and, in my view, leads to cleaner and more interpretable results. Is there a particular reason the authors chose not to use the projection formulation directly?***
>
> The main reason is that we found that the geometric objects - specifically the tangent space, the exponential map, and the logarithmic map - to be more easily describable in the quotient space representation compared to the projection matrix representation (see [BZA24] for a comparison). These objects appear in our results in Theorem 1 and Proposition 1, and play an even more prominent role in the proofs.
>
> A secondary reason is that we did not want there to be confusion between the vector space structure of matrices, and the Riemannian structure of the Grassmannian. For example, for two subspaces $[U]$, $[V]$, it is easy to point out that "$[U] - [V]$" is not naturally well-defined, and that $Log_{[U]}([V])$ can be thought of as a Riemannian replacement for this quantity, which we did in Section 2.1 in our description of the logarithmic map. This argument is more difficult to make when working with projection matrices, as their difference is well-defined from the vector space structure they inherit from the space of matrices.
>
>
> Of course, the downside of the quotient space formulation is that it is more abstract. Ultimately, this choice is subjective, and we thought the quotient space representation offered a better trade-off.
>
> We would be grateful for any suggestions the reviewer might have on how to further simplify the presentation.
>
> ***The results in Theorem 1 are not surprising. A closely related asymptotic result for the excess risk appears in Proposition 2.14 of Non-Asymptotic Upper Bounds for the Reconstruction Error of PCA by Reiss and Wahl (2020). Although their result is stated under a Gaussian assumption, the proof appears to use Gaussian primarily to ensure full-rank conditions and distinct eigenvalues, suggesting that the result extends beyond the Gaussian setting. The authors should clarify how their results differ in a substantive way from those in Reiss and Wahl (2020).***
>
> We thank the reviewer for pointing out the potential extension of Reiss and Wahl's result. We agree that, with some work, the argument behind Proposition 2.14 in Reiss and Wahl, under the eigengap assumption $\lambda_{k} > \lambda_{k+1}$ (otherwise the argument significantly depends on the Gaussian assumption - $\xi_{jk}$ are not independent in general), could be generalized, though the argument requires at least the existence of the fourth moments of $X$ as it requires the CLT on the sample covariance matrix.
>
> The excess risk result in Theorem 1 in our paper is proven using a completely different argument, and yields a slightly stronger result in that only finiteness of (8) is required - a weaker condition than the finiteness of the fourth moments of $X$. We will mention the above in our discussion of Theorem 1, in particular in the paragraph "Relationship with existing work and assumptions".
>
> We would like to emphasize that the asymptotic normality result in Theorem 1, to the best of our knowledge, has not appeared anywhere before in the literature. We found it interesting and somewhat surprising that the asymptotic normality happens on the tangent space of the minimizer.
>
> ***The authors do not provide numerical results to support their theoretical findings. Including a few synthetic experiments would help illustrate how to interpret the theoretical results and would also serve to validate their theoretical claims.***
>
> We will add a section containing results of simulations that compare our predictions with empirically measured quantities. Please note however that while such simulations help illustrate our results, they cannot validate them - the simulations can necessarily only be run on a handful of data distributions, whereas our claims are made for general data distributions with mild moment assumptions.

---

> > ### Comment · Reviewer_fztr · 2025-08-06
> >
> > Thank you for your response. I find the paper very interesting and appreciate the valuable tools it offers for low-rank subspace analysis. One minor point: the current presentation may be too dense for readers unfamiliar with Riemannian manifolds, especially given that many classical results in PCA were developed without relying on such tools. I encourage the authors to include more intuitive explanations that connect their results and key quantities to classical concepts in PCA and low-rank modeling. Doing so would make the paper more accessible to a broader audience. I have no further questions.

---

### Official Review · Reviewer_SW9G · 2025-07-01

**Clarity:** 2
**Significance:** 2
**Originality:** 2
**Rating:** 4
**Confidence:** 2

**Summary:**

This paper studies the excess risk of PCA under reconstruction loss and provides a precise asymptotic and non-asymptotic analysis. The authors establish a central limit theorem for PCA subspace estimation and derive the asymptotic distribution of excess risk. A matching non-asymptotic upper bound is also given. The key technical contribution is proving that the Rayleigh quotient is generalized self-concordant on the Grassmannian, enabling tight error control. The work theoretically characterizes how data distribution governs PCA’s excess risk.

**Questions:**

- How does your method behave in the absence of an eigengap (e.g., \lambda_k=\lambda_{k+1})?
- Can your framework be adapted to Kernel PCA via random feature methods or kernel tricks?
- The delta dependence in Theorem 2 is stated to be unimprovable, but it seems to be weaker. Can you compare previous work on assumption and delta dependence?

**Ethical Concerns:**

["NO or VERY MINOR ethics concerns only"]

**Final Justification:**

After clarifying from the authors, they moderately addressed my concerns so I've increased my score.

**Limitations:**

yes

**Paper Formatting Concerns:**

Lack limitations section

**Quality:**

3

**Strengths And Weaknesses:**

Strength:
- Establishes a new CLT for PCA subspace error and derives its asymptotic excess risk distribution.
- Proves generalized self-concordance of the Rayleigh quotient on the Grassmannian, a good geometric insight.

Weakness:
- Addresses a well-studied classical problem with incremental refinement rather than fundamentally new insights.
- It is not new that excess risk ~ 1/(n*eigengap) from Non-asymptotic upper bounds for the reconstruction error of PCA

---

> ### Author Rebuttal · Authors · 2025-07-31
>
> We thank the reviewer for reading and engaging with our paper. We address their questions and comments below.
>
> ***How does your method behave in the absence of an eigengap (e.g., $\lambda_{k}=\lambda_{k+1}$)?***
>
> Our analysis does not apply when $\lambda_{k} = \lambda_{k+1}$. As we discussed in lines 330-335, this is because in this case, there is infinitely many minimizers of the risk which form a submanifold of the Grassmannian, please see (14). The tools of asymptotic statistics that our analysis builds on do not apply to such a degenerate setting.
>
> We do believe that this is an interesting case to study, but we left it to future work.
>
> ***Can your framework be adapted to Kernel PCA via random feature methods or kernel tricks?***
>
> That's a really interesting question. If the RKHS is finite dimensional then our analysis applies. In the infinite dimensional case, the idea of approximating kernel PCA as a sequence of regular PCA on random features seems interesting. It's unclear to us how one could translate guarantees on PCA with random features to guarantees on the full Kernel PCA, but this seems like a potentially interesting approach.
>
> With the reviewer's approval, we would be happy to include this suggestion around line 346.
>
> ***The delta dependence in Theorem 2 is stated to be unimprovable, but it seems to be weaker. Can you compare previous work on assumption and delta dependence?***
>
> There has been a large body of work that shows that an inverse dependence on $\delta$ is inevitable for the excess risk of ERM procedures (of which PCA is an instance) (see e.g. [1, 2]). The sentence we wrote (lines 288-289) was perhaps too strong - If the reviewer agrees, we could replace it with: "The literature on the performance of ERM shows that, under weak moment assumptions, an inverse dependence on $\delta$ is inevitable (see e.g. [1, 2] and the survey [LM19]). This strongly suggests that the dependence on delta in the upper bound on the excess risk of PCA in Theorem 2 is unimprovable.".
>
>
> [1]: Lugosi, Gabor, and Shahar Mendelson. "Risk minimization by median-of-means tournaments." Journal of the European Mathematical Society 22.3 (2019): 925-965.
>
> [2]: Lecué, Guillaume, and Matthieu Lerasle. "Robust machine learning by median-of-means: theory and practice." (2020): 906-931.
>
> ***Addresses a well-studied classical problem with incremental refinement rather than fundamentally new insights.***
>
> We respectfully disagree that our work is incremental. Our work contains a fundamentally new insight - namely the self-concordance property of the block Rayleigh quotient over the Grassmannian. This was not known to the best of our knowledge, despite significant interest in this type of result as we mentioned in lines 276-279.
>
> In the paper, we have emphasized the application of this insight to the study of PCA, but we have also highlighted that it is relevant in studying eigenspace estimation in general (Remark 2), which appears in many important machine learning contexts (spectral clustering, community detection, contrastive learning, randNLA, etc). We have also mentioned the potential of using this self-concordance result to study the performance of first and second order optimization algorithms on the block Rayleigh quotient (lines 357-260), as studied in [AV24] for example.
>
> If the reviewer believes that our writing didn't emphasize the importance of the self-concordance result enough, we are open to suggestions on how to make this clearer in the presentation.
>
> ***It is not new that excess risk ~1/(n * eigengap) from "Non-asymptotic upper bounds for the reconstruction error of PCA".***
>
> Please correct us if we misunderstood your comment, but the statement that the excess risk is upper and lower bounded by 1/(n*eigengap) is in general false. Asymptotically, what we have shown (and to the best of our knowledge we are the first to do so) is that that the excess risk tightly concentrates around
> $$
> \frac{1}{n}\sum_{i=1}^{d-k}\sum_{j=1}^{k} \dfrac{\mathbb{E}\left[\langle u_{k+i}, X \rangle^{2} \langle u_j, X \rangle^{2} \right]}{\lambda_j - \lambda_{k+i}},
> $$
> under very mild moment assumptions on the data distribution (see Corollary 1). Additionally, Theorem 2 provides a non-asymptotic upper bound on the excess risk that, up to an absolute constant, exactly recovers this term.
>
> The non-asymptotic upper bounds found in Reiss and Wahl 2020 [RW20] only hold for subgaussian distributions, and do not in general recover this term - please see the discussion at the top of page 1110 in their manuscript.
>
> If we misunderstood the reviewer's comment, we'd be happy to discuss further.

---

### Official Review · Reviewer_VhMx · 2025-07-01

**Clarity:** 4
**Significance:** 4
**Originality:** 4
**Rating:** 5
**Confidence:** 5

**Summary:**

This paper provides a precise theoretical characterization of Principal Component Analysis (PCA) performance. Two main key contributions of the paper include: (1) a central limit theorem for the error of the principal subspace estimated by PCA, with the asymptotic distribution of excess risk; and (2) a non-asymptotic upper bound on the excess risk that recovers the asymptotic characterization in the large-sample limit. The analysis is conducted on the Grassmannian manifold, thus ensuring the usual orthogonality constraints on the principal components.

**Questions:**

If the authors could provide satisfactory answers to the following questions, it would mean the work potentially has more impact in statistics (and in particular, the analysis of identifiable models) than what is already discussed in the paper.

1) We know traditional PCA can be seen as the solution to the maximum likelihood estimation of the Probabilistic PCA (PPCA) model when the variance of the noise term tends to zero. I was wondering what could be a potential connection of this work to an already published work: "On the Consistency of Maximum Likelihood Estimation of Probabilistic Principal Component Analysis", NeurIPS 2023, which does its analysis in a similar quotient space framework. In particular, is the current work helpful for extending the results discussed in NeurIPS '23 to answer questions around the rate of convergence and asymptotic efficiency of the MLE of the PPCA model? A discussion around this would be insightful, as I believe these models and approaches are connected.

2) What about extending your results to general M-estimators of statistical models that include identifiability? I understand we can get rid of identifiability by considering an appropriate quotient of the parameter space. Let us also assume the quotient parameter space is a Riemannian manifold (though often that's not the case). Is there potential that your results generalize in such cases? I think proving a result like Corollary 2 might be challenging in all generality, but I would like to know the authors’ insights on this.

**Ethical Concerns:**

["NO or VERY MINOR ethics concerns only"]

**Final Justification:**

I am satisfied with author's response.

**Limitations:**

yes

**Quality:**

4

**Strengths And Weaknesses:**

Strengths:

The work analyzes PCA as an M-estimator on the Grassmannian manifold. The paper provides both asymptotic (Theorem 1) and non-asymptotic (Theorem 2) results, with the non-asymptotic bounds recovering the asymptotic behavior. Such results are of interest/useful to both theoreticians/practitioners. Some of the key technical contributions include: the proof that the reconstruction risk is generalized self-concordant (Proposition 1) and the explicit formulas for the excess risk in terms of the data distribution's fourth moments and eigenvalue gaps (Equation 21). The authors provide a discussion around how their work is applicable to generalized eigenvalue problems (Remark 2), with applications to spectral clustering, community detection, and contrastive learning. Despite the technical complexity, the paper is well-written with good intuition provided throughout and overall an enjoyable read for an expert working in PCA.

Weaknesses:

The math in the paper is elegantly explained and I do not have any major objections. Obvious limitations are properly addressed (i.e., the eigengap assumption and the finitude of the fourth moment are usual assumptions, ubiquitous in the literature and not really a major bottleneck in my opinion). Theorem 1 could be explained with more intuition. Could you please explain why the logarithm naturally appears in the context? Apart from this, I think the paper could also benefit from a bit more discussion around its connection to relevant literature (please see questions).

---

> ### Author Rebuttal · Authors · 2025-07-31
>
> We thank the reviewer for engaging with our paper and for their positive comments. We address their questions below.
>
> ***Theorem 1 could be explained with more intuition. Could you please explain why the logarithm naturally appears in the context?***
>
> In general, $Log_{[U]}([V])$ can be viewed as the rigorous, Riemannian analogue of "$[V] - [U]$". Comparing Theorem 1 with the classical mutlivariate CLT, the vector $Log_{[U_\*]}([U_{n}])$ can thus be interpreted as the Riemannian analogue of the vector $\mu_n - \mu_{\*}$ that takes one from the true mean $\mu_{\*}$ to the sample mean $\mu_n$.
>
> ***We know traditional PCA can be seen as the solution to the maximum likelihood estimation of the Probabilistic PCA (PPCA) model when the variance of the noise term tends to zero. I was wondering what could be a potential connection of this work to an already published work: "On the Consistency of Maximum Likelihood Estimation of Probabilistic Principal Component Analysis", NeurIPS 2023, which does its analysis in a similar quotient space framework. In particular, is the current work helpful for extending the results discussed in NeurIPS '23 to answer questions around the rate of convergence and asymptotic efficiency of the MLE of the PPCA model? A discussion around this would be insightful, as I believe these models and approaches are connected.***
>
> We thank the reviewer for pointing out this interesting connection.
>
> We don't believe that the specific results obtained in our work are directly applicable to the PPCA model, though the general approach we took could potentially be adapted for PPCA.
>
> Indeed, in the case of PCA, the search space is the Grassmannian, which can equivalently be viewed as the quotient space $\mathbb{R}\_{\*}^{d \times k}/GL(k)$ where $\mathbb{R}\_{\*}^{d \times k}$ is the set of full rank $d \times k$ matrices. The search space for PPCA seems to be different: for the parameter $W$ we believe it is $\mathbb{R}_{*}^{d \times k} / O(k)$. If this set can be made into a Riemannian manifold (we believe it can, but didn't check carefully), and if one can prove a version of Corollary 2 for the negative log-likelihood under this Riemannian structure, then one can in principle obtain results similar to Theorems 1 and 2 for PPCA.
>
> One difficulty in carrying out this program is that the exponential map on $\mathbb{R}_{*}^{d \times k} / O(k)$ likely doesn't admit a simple analytical form, which would make proving a statement of the form of Corollary 2 difficult for the case of PPCA. On the other hand, in PPCA we know that the data is Gaussian, which can potentially significantly ease the analysis.
>
> ***What about extending your results to general M-estimators of statistical models that include identifiability? I understand we can get rid of identifiability by considering an appropriate quotient of the parameter space. Let us also assume the quotient parameter space is a Riemannian manifold (though often that's not the case). Is there potential that your results generalize in such cases? I think proving a result like Corollary 2 might be challenging in all generality, but I would like to know the authors’ insights on this.***
>
> It seems plausible that a theory of M-estimation, analogous to the one developed in [Van00] (Chapter 5) for Euclidean spaces, could be developed in the more general setting of Riemannian manifolds (with perhaps some additional regularity). This could potentially yield general statements from which Theorem 1 could be deduced.
>
> The non-asymptotic theory is more subtle - the main challenge is to get explicit control on the error of the second order Taylor expansion of the negative log likelihood around its minimizer, and this is typically done on a case by case basis. (Generalized) self-concordance has proven to be a powerful tool to obtain such control. Not all negative log likelihoods satisfy it, but there has been work showing that it holds for many important regression problems in Euclidean settings [OB21]. A main contribution of our work is showing that it holds for PCA, formulated as an M-estimation procedure on a Riemannian manifold.
>
> We don't know whether this self-concordance property holds for a broader set of problems of interest. We don't expect it to always hold, but there is hope that it holds for some other M-estimation procedures of interest on Riemannian manifolds.

---

> > ### Comment · Reviewer_VhMx · 2025-08-05
> > **Thank you**
> >
> > Thank you for the detailed response. I maintain my positive outlook on this paper.

---

### Decision · Program_Chairs · 2025-09-17

**Decision:**

Accept (poster)

**Comment:**

This paper provides a rigorous theoretical study of PCA’s excess risk under reconstruction loss. The contributions include (1) a new CLT for PCA subspace error with an explicit asymptotic excess risk distribution, and (2) a matching non-asymptotic upper bound derived via self-concordance of the block Rayleigh quotient on the Grassmannian. Reviewers appreciated the technical depth, solid proofs, and clear connection to eigenspace estimation problems, and one reviewer strongly recommended acceptance.

Concerns centered on the degree of novelty relative to prior work (e.g., Reiss & Wahl 2020), the lack of empirical illustrations, and the density of exposition for readers unfamiliar with Riemannian geometry. The rebuttal clarified differences from prior results, emphasized the new self-concordance insight, and committed to adding simulations and clearer explanations. While some reviewers still view the results as incremental, the consensus is that the theoretical contribution is technically solid, broadly relevant, and above the bar.

I recommend accept the paper.